# Scalable high yield exfoliation for monolayer nanosheets

Zhuyuan Wang[1], Xue Yan[2], Qinfu Hou [1], Yue Liu[1], Xiangkang Zeng [3], Yuan Kang[1], Wang Zhao[1], Xuefeng Li[1], Shi Yuan[1], Ruosang Qiu[1], Md Hemayet Uddin[4], Ruoxin Wang[1], Yun Xia[1], Meipeng Jian [1], Yan Kang[5], Li Gao[6], Songmiao Liang[5], Jefferson Zhe Liu [2], Huanting Wang [1] & Xiwang Zhang [1,3] ✉

Although two-dimensional (2D) materials have grown into an extended family that accommodates hundreds of members and have demonstrated promising advantages in many fields, their practical applications are still hindered by the lack of scalable high-yield production of monolayer products. Here, we show that scalable production of monolayer nanosheets can be achieved by a facile ball-milling exfoliation method with the assistance of viscous poly-ethyleneimine (PEI) liquid. As a demonstration, graphite is effectively exfoliated into graphene nanosheets, achieving a high monolayer percentage of 97.9% at a yield of 78.3%. The universality of this technique is also proven by successfully exfoliating other types of representative layered materials with different structures, such as carbon nitride, covalent organic framework, zeolitic imidazolate framework and hexagonal boron nitride. This scalable exfoliation technique for monolayer nanosheets could catalyze the synthesis and industrialization of 2D nanosheet materials.

Scalable production of monolayer 2D materials with minimized defects and much preserved pristine properties of their bulk counterparts are critical in many applications[1-3]. Although a number of monolayer 2D nanosheets have been achieved via bottom-up synthesis and chemical exfoliation methods, they are either restricted to a limited scale or introducing a large number of defects that impeded their applications[4-7]. Due to great scalability and mild processing conditions, mechanical exfoliation shows the potential for achieving a scalable production of high-quality 2D nanosheets. Plenty of mechanical exfoliation techniques, e.g., ultrasonication[8,9], high-speed mixing[10], and ball milling[11], have been explored for the exfoliation of layered materials in liquid medium. Nanosheet products of these exfoliation methods, however, are predominantly thick multi-layer nanoflakes. The monolayer-nanosheet percentage in these products is often less than ~20%[3,8,10].

The failure of producing monolayer nanosheets at high yields by current liquid phase mechanical exfoliation (LPE) is thought to be the result of insufficient delamination force applied on layered materials[12]. It has been proven that simply enhancing shear rate (e.g., increasing sonication intensity or mixing rate) unfortunately has limited improvements on exfoliation in term of yield and monolayer percentage[13,14]. To achieve effective exfoliation, shear force in liquids must be effectively transferred to the targeted materials. Nonetheless, the force transfer in current LPE methods is insufficient because of the low viscosity of commonly used solvents[10,15]. More recently, dry-state ball-milling using solid diluents has been explored as an alternative to LPE. Although enhanced exfoliation yield was achieved due to the high force transfer efficiency of solid diluents, the obtained 2D nanosheet products possess broad thickness distribution and small lateral size

[1]Department of Chemical and Biological Engineering, Monash University, Clayton, VIC 3800, Australia. [2]Department of Mechanical Engineering, The University of Melbourne, Parkville, VIC 3010, Australia. [3]UQ Dow Centre for Sustainable Engineering Innovation, School of Chemical Engineering, The University of Queensland, St Lucia, QLD 4072, Australia. [4]Melbourne Centre for Nanofabrication, 151 Wellington Road, Clayton, VIC 3168, Australia. [5]Vontron Membrane Technology Co. Ltd., No. 1518 Liyang Road, Guiyang, Guizhou 550014, People's Republic of China. [6]South East Water Corporation, PO Box 2268 Seaford, VIC 3198, Australia. ✉e-mail: xiwang.zhang@uq.edu.au

due to intensive crushing coupled with shearing during dry-state mixing[16,17].

Inspired by the well-known scotch-tape exfoliation[18–20], herein, we introduce a technique called sticky mechanical exfoliation by applying liquid polymer as an exfoliation medium in ball-milling method to achieve a scalable production of monolayer nanosheets based on two main interlocked criteria: (1) Exfoliation medium should possess a high viscosity to facilitate shear force transmission; (2) Exfoliation medium should have a strong adsorption energy on common layered materials to stick on the layered material during delamination process. Preliminary screening experiments shows that high-viscosity, branched polyethyleneimine (PEI) is an appropriate candidate. With the assistance of PEI, graphite and other four types of representative layered materials with different structures including, carbon nitride, covalent organic framework, zeolitic imidazolate framework, and hexagonal boron nitride, are effectively exfoliated into monolayer nanosheets at high yields.

## Results and discussion
### Exfoliation of graphite
The feasibility of this technique was first exemplified by graphite exfoliation. In a typical operation, graphite and PEI were mixed in milling jars and milled at 500 rpm for 5–15 h, and exfoliated products were collected and dispersed in water. The dispersion was then repeatedly rinsed with water on a homemade filtration device to remove free PEI molecules (Supplementary Fig. 1.2), followed by

centrifugation to separate graphene nanosheets from unexfoliated graphite. Apparent yields were calculated to be from 52.4 to 90.7% by weighing the solid content in graphene dispersion products. The crystalline structure of graphite is well preserved after this mechanical exfoliation, suggested by powder X-ray diffraction patterns (p-XRD) (Fig. 1b)[21]. The graphene powder samples (freeze-dried from rinsed water solution) show broad peaks (100)/(101) at 43–45° and (110) at ~76°, corresponding to the 2D in-plane symmetry along graphene sheet. The decreased intensity of these peaks indicates the decreased lateral size with prolonging the milling time. All powder samples show a weak peak (002) at ~26°, a characteristic of paralleled graphene layers, which could mainly arise from nanosheet re-stacking during sample drying process.

When rinsed graphene solution was deposited on mica, plenty of small flat disks with a lateral size of hundreds of nanometers and a height between 0.5 to 1 nm were observed (Fig. 1d–f). In addition, their apparent heights exhibit a narrow unimodal distribution with deviations less than 0.2 nm (Supplementary Figs. 2.1–2.3). These AFM characteristics suggest that these observed sub-nm disks may be monolayer graphene[22,23]. To confirm the finding, we collected accumulated Raman signals from these disks on mica (Supplementary Fig. 2.5). The generated patterns feature characteristic D, G, and 2D bands of graphene. The intensity ratios of $I_{2D}/I_G$ locate in the range of 1.06 to 3.1, implying high monolayer percentage in the spotted graphene[8] (Fig. 1c and Supplementary Fig. 2.5). In addition, the intensity ratio of $I_D/I_G$ band, which is widely used as an indicator of sp$^3$

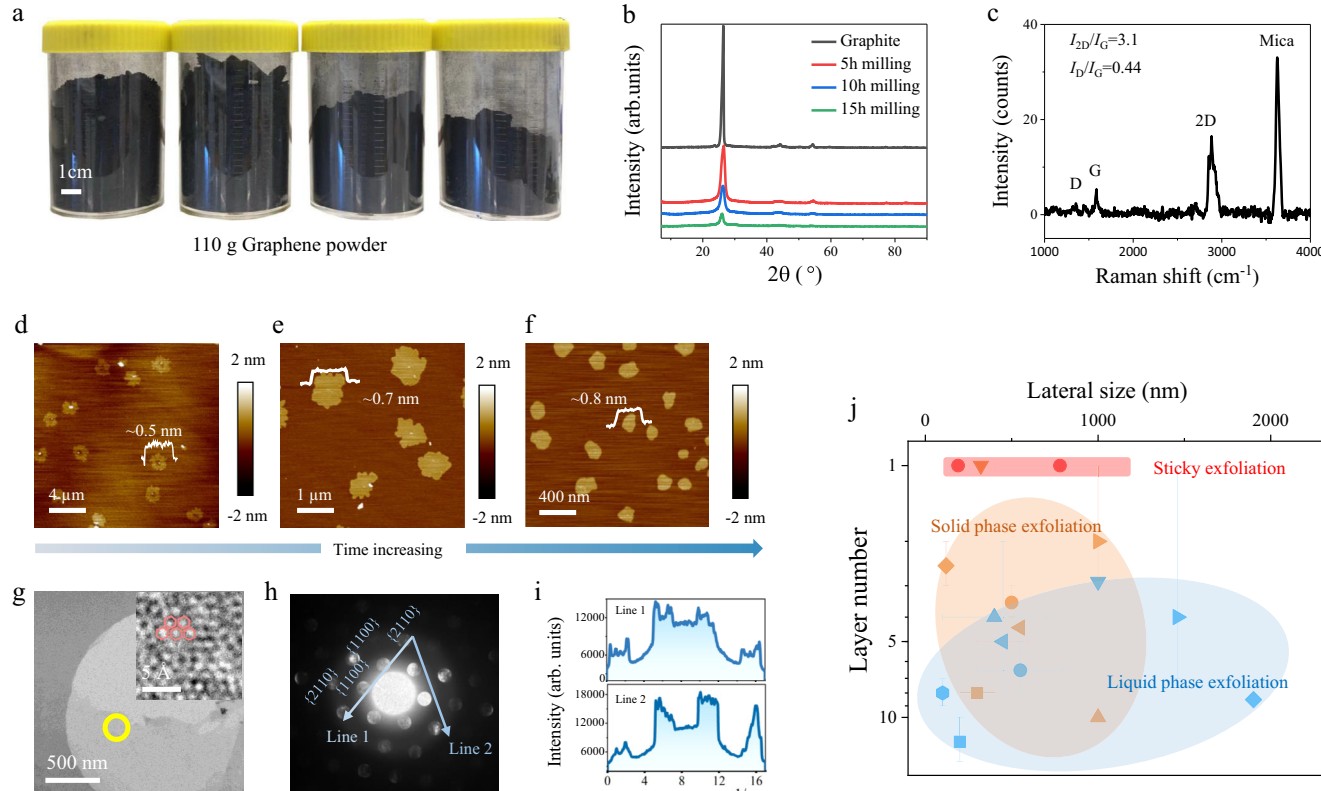

**Fig. 1 | The physical properties of the graphene nanosheets via sticky mechanical exfoliation. a** Photograph of the obtained 110 g graphene in powder form. **b** Powder X-ray powder diffraction (XRD) patterns of pristine graphite and graphene products after 5, 10, and 15 h milling. **c** Accumulated Raman spectrum of the graphene nanosheets after 15 h milling deposited on mica. Detailed characterization method can be found in supplementary S2.4. **d–f** Selected atomic force microscopy (AFM) height profiles of graphene exfoliated by 5, 10, and 15 h milling, respectively (from left to right). **g** High resolution transmission electron microscopy (HR-TEM) image of exfoliated graphene (inset, atomic resolution).

Red hexagons highlight the graphene lattice. **h** Converged beam electric diffraction (CBED) of graphene indicated by the yellow circle in (**e**). Spot size: 9, convergence angle: ~15 milliard. **i** Diffraction spot intensity taken along the lines in (**f**). **j** Comparison of the obtained graphene in this work with other mechanical exfoliated graphene in terms of lateral size and layer number. Error balls indicate the data range provided in the references. (Detailed data in Supplementary Table 2.1.) Blue dots: liquid phase exfoliation; Orange dots: solid phase exfoliation; Red dots: this work.

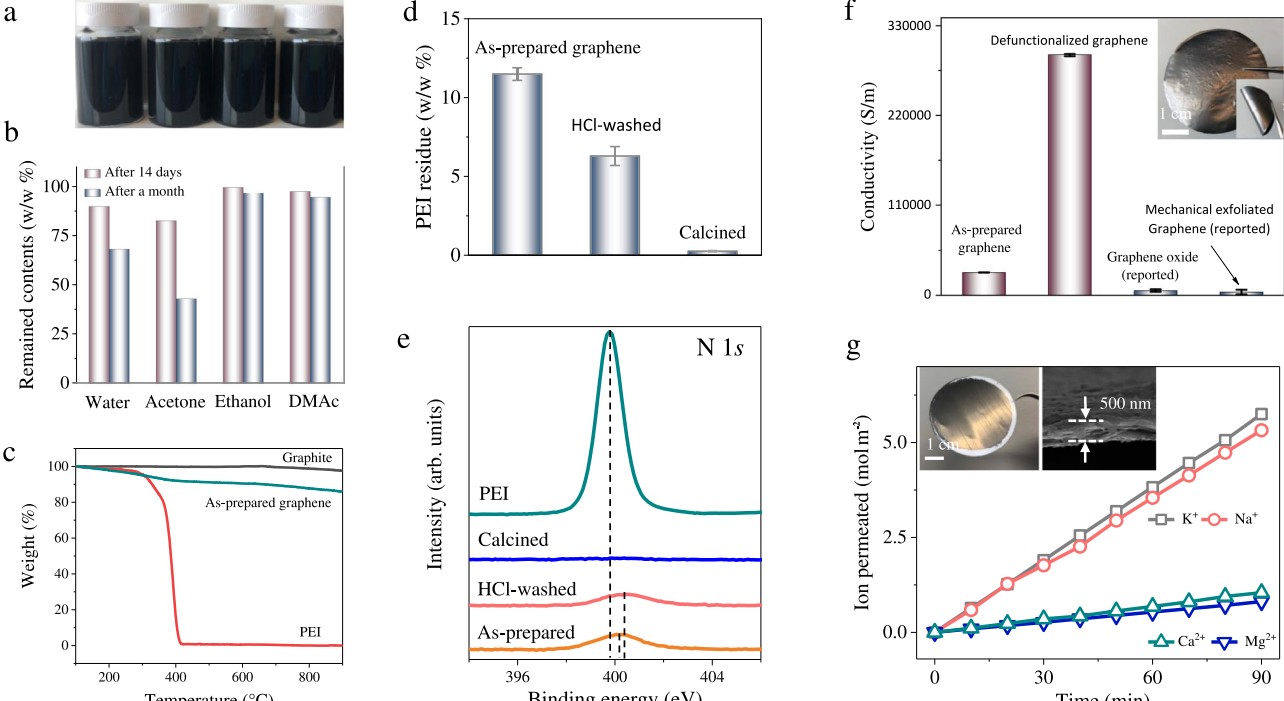

**Fig. 2 | Properties and potential applications of the exfoliated graphene nanosheets. a** Photograph of graphene nanosheets (10 h milling) dispersed in solvents including water, ethanol, isopropanol (IPA) and dimethylacetamide (DMAc) at a concentration of 0.5 mg/mL. **b** The weight ratio of remained graphene nanosheets in these solutions after fortnight and 30 days, respectively. **c** Thermogravimetric analysis (TGA) of graphite, polyethyleneimine (PEI) and exfoliated graphene (10 h milling), respectively. **d** Content of residual PEI on as-prepared graphene (10 h milling) and de-functionalized graphene by acid washing and incineration under N₂ atmosphere. **e** High-resolution X-ray photoelectron spectroscopy (XPS) spectra of N 1s of PEI, as-prepared graphene and de-functionalized graphene. The dashed lines highlight the peak shifts. In pure PEI, a sharp peak at 399.8 eV was recognized as typical groups of -N/NH/NH₂ in PEI.

A weak and broad N 1s peak also appears in exfoliated graphene but with a -0.4 eV positive shift which might be caused by electron transfer from the polymer chains of PEI to graphene[41]. After HCl-treatment, the N 1s peak tends to be weaker and has more shift in comparison to pristine graphene, meaning fewer nitrogen remained and greater electron transfer between PEI and graphene. **f** The conductivity of the as-prepared graphene and defunctionalized graphene in comparison with reported graphene oxide and sonication exfoliated graphene. Inset is the photograph of the graphene film. **g** The permeation rate of monovalent ions (K⁺ and Na⁺) and divalent ions (Mg²⁺ and Ca²⁺) as a function of diffusion time in laminar membrane assembled using graphene nanosheets, inset are the photograph and scanning electron microscopy (SEM) cross-section image of the graphene membrane. All error bars indicate the standard deviation of the experiments.

carbon defects, is between 0.38 and 0.44 for our graphene samples. The $I_D/I_G$ ratio is lower than those of graphene oxide (GO) and reduced GO (rGO) (-0.7–0.9)[24], indicating the much preserved high-quality lattice of the as-prepared graphene nanosheets. Transmission electron microscopy (TEM) characterization also shows that these graphene nanosheets have lateral sizes of hundreds of nanometers, in agreement with AFM characterization. The weak contrast of these nanosheets to TEM carbon grid reveals that they are ultrathin. As illustrated in Fig. 1g–i and Supplementary Fig. 2.6, the TEM diffraction patterns display a typical six-fold symmetry and characteristic fingerprint of monolayer graphene with more intense inner spots ({1010} facets) than outer ones ({2110} facets)[8,10,25], providing further evidences that the as-exfoliated products are dominated by monolayer graphene.

A statistical analysis on the AFM height profiles of over 200 graphene nanosheets (after 10 h milling) shows that 91.2% of the nanosheets are monolayered and their lateral sizes are averaged at 780 nm (Fig. 1e and Supplementary Fig. 2.1). More importantly, the lateral size of exfoliated graphene nanosheets can be tailored by controlling milling time. By extending milling time from 10 to 15 h, the average lateral size of graphene nanosheets decreases from 780 to 190 nm with an apparent yield of 90.7% (Supplementary Fig. 2.4). The percentage of monolayer graphene further increases to 97.9% according to a statistical analysis towards over 500 nanosheets (Fig. 1f and Supplementary Fig. 2.2). By contrast, the averaged lateral size of graphene nanosheets rises to 1.64 μm when milling time is reduced to 5 h, albeit with a slightly reduced monolayer percentage of 76% (Fig. 1d

and Supplementary Fig. 2.3), which is still much higher than those of other mechanical exfoliation methods (Fig. 1j and Supplementary Table 2.1). By enlarging milling jars, increasing graphite addition, and optimizing milling parameters, more than 40 g of graphene was produced in one batch using a small lab mill. Intriguingly, apart from a consistent high monolayer percentage, reduced PEI consumption comes as a bonus when the production is scaled up (Supplementary Section 3.1).

### Solvent dispersibility of as-prepared graphene
The as-prepared graphene nanosheets have a good dispersibility in both water and a number of organic solvents, e.g., ethanol, acetone, and dimethylacetamide (DMAc) (Fig. 2a, b and Supplementary Fig. 3.2). Graphene nanosheets remain stably dispersed in these solutions even after one month of standstill, in particular in ethanol and DMAc. The graphene water dispersion can be freeze-dried into powder and re-dispersed into water–ethanol mixture through 30 min sonication. Only -17% of graphene nanosheets is lost in the drying-redispersion process due to re-stacking (Supplementary Section 3.3). Compared to most reported graphene products that have to be stored in solutions due to poor re-dispersibility after drying[10], the excellent re-dispersibility of the graphene nanosheets synthesized via this sticky milling offers crucial convenience in practical transport, storage, and processing. We believe that the excellent re-dispersibility of the graphene nanosheets is attributed to a small amount of residual PEI on graphene nanosheets. Elemental analysis, X-ray photoelectron spectroscopy (XPS) and

thermogravimetric analysis (TGA) analysis revealed that the weight percentage of residual PEI on the graphene nanosheets is in the range of 6.3–12.4 wt% (Fig. 2c–e and Supplementary Sections 4.1, 4.2). The residual PEI molecules can be partially removed via acid washing or completely removed via incineration at 600 °C under $N_2$ atmosphere (Fig. 2d, e and Supplementary Section 4.2).

## Reassembling exfoliated graphene into hierarchical structures

To demonstrate the high quality of exfoliated graphene nanosheets for potential applications, a flexible free-standing film with 18.2-μm thickness was fabricated. The film exhibited a low sheet resistance of 1.93 $\Omega/sq^{-1}$, corresponding to a conductivity of ~28,000 S/m. The conductivity was further promoted to ~322,000 S/m by removing residual PEI (Fig. 2f, Supplementary Section 4.3), much higher than those of GO (4000–7000 S/m)[26–28] and other mechanical exfoliated graphene (1200–6500 S/m)[8,29,30]. In addition, due to the presence of residual PEI, the as-prepared graphene nanosheets are highly positively charged with a zeta potential of up to +52.9 mV at pH of ~6 (Supplementary Section 4.4), distinguishing them from other reported graphene nanosheets that are often negatively charged in neutral solution. To highlight their charge property and high aspect ratio, graphene nanosheets were parallelly piled into a membrane with laminar channels (Fig. 2g inset). The laminar membrane showed an ultra-fast transport of monovalent ions ($Na^+$, $K^+$) at a rate up to 3.8 mol m$^{-2}$ h$^{-1}$ while a relatively low transport rate of divalent ions ($Ca^{2+}$, $Mg^{2+}$), leading to a competitive mono/divalent ion selectivity of 5.11–7.0 (Fig. 2g). An apparent drop in mono/divalent selectivity was observed after HCl treatment on the membrane due to partial removal of PEI molecules, which also illustrates the role of PEI in ion transport (Supplementary Fig. 4.3). This ion transport behavior in positive-charge governed nanoconfinement could be leveraged for ion-separation, energy harvesting and storage, and sensing technologies[31–35].

## Exfoliation mechanism

To understand the exfoliation mechanism of this sticky ball milling method, we first compared the geometry of intermediate products after 1 and 2 h-milling (Fig. 3a). AFM topographic images show that both thin-yet-large and thick-yet-small nanosheets exist in 1 h exfoliation product (Fig. 3a and Supplementary Fig. 5.1). The distributions of thickness and lateral size gradually narrow down with milling time extended to 2 h. The results indicate that graphite particles undergo breaking and delamination simultaneously during the milling process. A partial exfoliated graphene shows height steps of ~0.7 nm (Supplementary Fig. 5.2), which is close to the apparent thickness of mono-layer graphene measured by AFM (Fig. 1d–f). To elucidate the role of grinding balls and PEI in exfoliation, discrete element method (DEM) was then employed to study their movement and interactions. The results suggest that besides collective orbital revolution around mill axis, grinding balls roll and slide against each other, thus generating normal colliding force and tangential sliding relative velocity as breaking and delamination driving forces, respectively (Fig. 3c, d and Supplementary Section 5.2).

The in-plane breaking of graphite occurs when sufficient compression force is applied following the random collisions between grinding balls. Statistical analysis on this force shows that the compression force is in the range of 0 to 2 N (Fig. 3c), which is too low to reach the graphene breaking threshold (125 GPa)[36] if ball-graphite contact surface is ideally smooth. However, a microscale observation on grinding ball surface reveals a ridge-and-valley morphology with an average height of 0.39 μm, and the distance between neighboring tips is averaged at 2.92 μm (Fig. 3b and Supplementary Fig. 5.6). The sharp ridges concentrate the pressure on graphite particles, which tremendously reduces the required breaking force to only 0.125 N if simplifying these tips into a cone shape (Fig. 3f and Supplementary

Section 5.2). In sticky exfoliation, PEI molecules will fill in these valleys and partially cover the rough ball surface due to their intrinsic stickiness and flexibility[37] (Fig. 3g). PEI macromolecular matrix can thus act as a buffer layer between balls and graphene and thus moderate the compression forces in terms of their magnitude and direction to avoid excessive breaking. This is further evidenced by that only carbon nanoparticles were produced in the absence of PEI (Supplementary Fig. 5.7). Since the in-plane breaking is primarily because of the protruding tips on the surface of grinding balls, the randomly distributed tips lead to a random in-pane breaking in terms of directions and locations on nanosheets accordingly. This random breaking process differs from the typical LPE exfoliation processes in which breaking occurs along the crystalline structure deformation directions, mostly along zigzag direction. As a result, the obtained nanosheets (Fig. 1d–f) as well as other nanosheets from ball milling processes[17,38] tend to show more irregular edges (serrated) than that of nanosheets from sonication exfoliation strategies with sharp edges[8,9].

Meanwhile, the delamination of graphite is enabled by the relative sliding of neighboring grinding balls in the presence of PEI. This is primarily because that highly viscous PEI establishes a velocity gradient perpendicular to the sliding direction, leading to fast and slow flowing rate near its interface with balls and graphite, respectively (Fig. 3g). Since PEI holds graphite firmly with an adsorption energy (−75.7 meV/unit area, Supplementary Table 5.1) 4.3 times higher than the interlayer binding energy of graphite (−17.4 meV/unit area), this flow rate difference ($\triangle V$) can be transferred to strong shear force to enable the sliding of neighboring layers. Interestingly, the required shearing force for the dislocation of adjacent layers is highly direction-dependent. The preferential sliding direction is the zigzag direction with a maximum required shearing strength of 0.116 GPa, which is only a third of the required shearing strength (0.342 GPa) when sliding along the armchair direction (Fig. 3e and Supplementary Figs. 5.8, 5.9). Since the direction of the generated shear force by grinding balls relative to the graphite lattice is completely random, sliding is energetically easy to be trigged whenever the shear force direction coincides with the zigzag direction. Moreover, upon the dislocation, newly exposed graphene will be covered by PEI molecules driven by the high adsorption energy, thus preventing its restacking[17,39].

A simplified two-plane model can be developed to quantitatively corroborate the sliding exfoliation (Fig. 3g). Based on the model, a critical viscosity of PEI liquid is required to be around 1005 mPa·s for graphite exfoliation under the experimental conditions of this study. However, our control tests using PEI liquid with different viscosity (from 1508 to 150,882 mPa·s, at 20 °C) found that the viscosity of PEI liquid needs to be over 7235 mPa·s to obtain monolayered graphene in exfoliation products (Supplementary Section 5.6). This inconsistency is though not unreasonable considering the negative correlation between temperature and viscosity of PEI. Infrared thermal imaging shows that PEI in mill jar was heated to around 50 °C owing to the exothermic milling process (Supplementary Fig. 5.11B). It thereby reduces the in-situ PEI viscosity by 7 times to around 1068 mPa·s, agreeing well with the predicted value based on the two-plane model (Supplementary Section 5.6). The failure of obtaining monolayer graphene nanosheets when using low-viscosity PEI also excludes molecular intercalation as an alternative exfoliation mechanism, which is energetically possible concerning the high adsorption energy.

## Universality of this exfoliation method

Furthermore, the universality of this sticky technique was assessed from both theoretical and experimental perspectives. To diversify the methodology, four different layered materials were selected as targets of interest, including materials weakly held via van del Waals forces and/or π–π interaction (covalent organic framework TAPB-PDA (COF), zeolitic imidazolate framework (ZIF-L), porous graphitic carbon nitride ($g-C_3N_4$)), and materials strongly held via intense polarized interaction

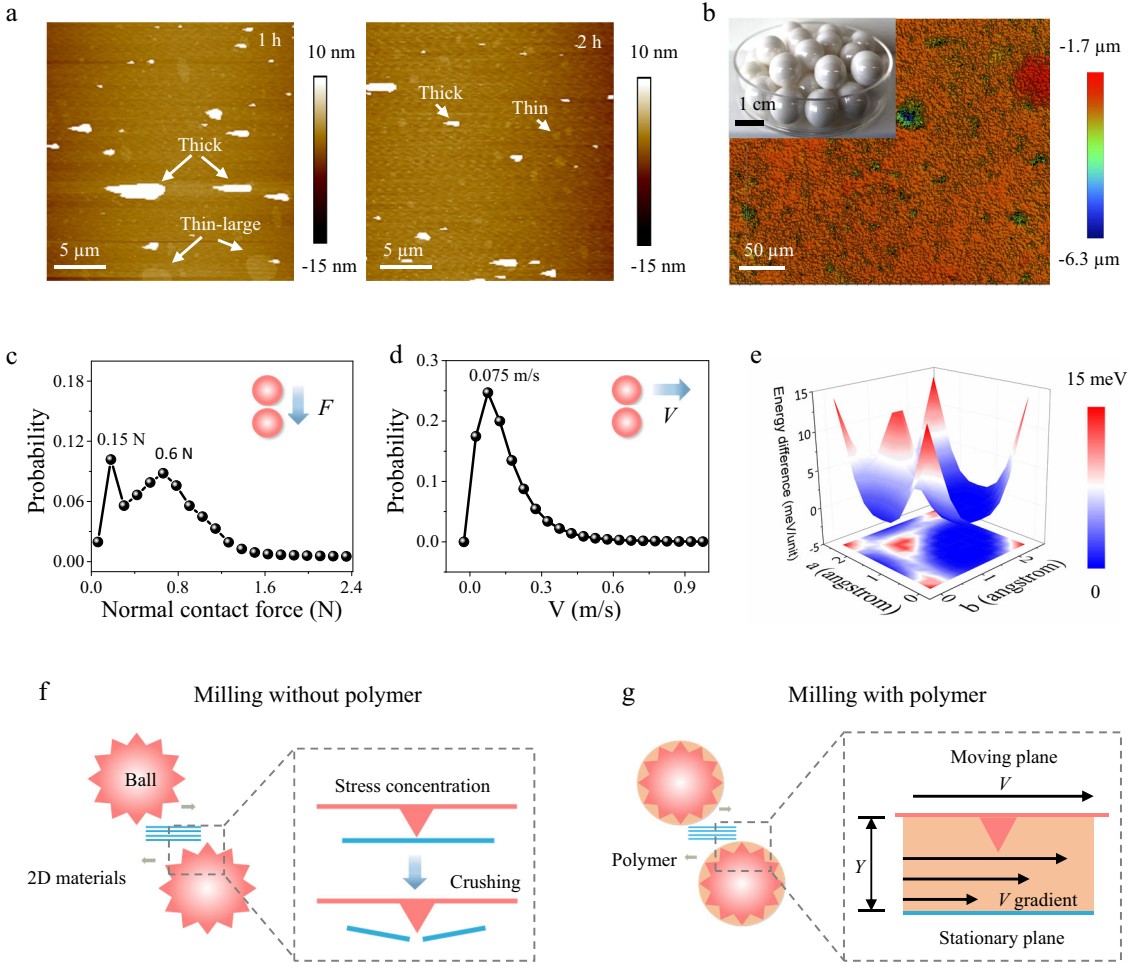

**Fig. 3 | Exfoliation mechanism of the sticky exfoliation. a** AFM characterization on graphene products obtained after 1 and 2 h milling, respectively. **b** Surface roughness of the grinding balls ($d = 1$ mm) characterized by an optical profilometer. Inset is a photograph of $ZrO_2$ grinding balls ($d = 10$ mm). **c** The statistical analysis of the normal compression force, and **d** the relative velocity of motions between grinding balls during ball milling process obtained by discrete element method (DEM) simulation study. **e** Density-functional theory (DFT) three-dimensional potential energy surface for the sliding motion of bilayer graphene. This figure was obtained by statically analyzing DFT energy difference relative to the ground state when sliding a bilayer graphene. *x* and *y* indicate the moving displacement along lattice vector direction in graphene unit. *z*-axis represents energy difference. The greater energy difference gradient means the harder sliding. The sliding direction

with the lowest energy difference gradient is considered as the preferential direction and the energy difference gradient along the preferential direction is taken as the minimum required shearing strength for delamination. **f, g** Two-plane model for describing the exfoliation process when milling with and without polymer. In this model, a moving plane (grinding balls) is topped on a fixed plane (layered materials, area = $A$), but separated by a polymer buffer layer (thickness = $Y$). To initiate the dislocation, shear force applied by the polymer ($F$) needs to exceed the required sliding force ($P \times A$) between two graphene layers, as represent in equation (1): $F = \rho \frac{\triangle V}{Y} A > P \times A$ (1). where $P$ is sliding stress (maximal 0.116 GPa by DFT calculation, Fig. 3e, Supplementary Table 5.2), and $\rho$ is the dynamic viscosity of the polymer, $\triangle V$ is the flow rate difference in the buffer layer, $\frac{\triangle V}{Y}$ is the velocity ($V$) gradient.

(hexagonal boron nitride (h-BN)). DFT calculation indicates the applicability of this sticky milling to the exfoliation of these materials from two key aspects. On one hand, the highest shearing strength along the preferential sliding direction of these materials still locates at the same magnitude as that of graphene, ranging from 0.01 to 0.190 GPa and following the sequence of COF < Graphene < g-$C_3N_4$ < h-BN (Fig. 4b and Supplementary Table 5.2). On the other hand, PEI shows a strong adsorption to these materials with absorption energy at level of 3.3 to 742.9 times of their interlayer binding energy (Fig. 4c and Supplementary Table 5.1). With these two prerequisites being fulfilled, PEI is theoretically able to transfer the shear force generated between grinding balls to the layered targets and lead to effective exfoliation.

Apart from theoretical study, the universality of this method was further supported by experimental exploration. With the same experimental setup and parameters used in graphene exfoliation, g-$C_3N_4$, ZIF-L and COF products achieved a satisfactory monolayer percentage of over 85% while maintaining an average lateral dimension

of 275 nm, 1.2 μm and 166 nm, respectively (Fig. 4a, d–i and Supplementary Section 6.1). However, only 5% of h-BN nanosheets were monolayered in the initial trial as a likely result of stronger interlayer attraction. In addition to van del Waals force, two adjacent h-BN layers are also held by partially iconic B-N bond (lip–lip interactions)[40], which requires 74% higher shear strength than graphene layers according to DFT calculations (Fig. 4b). In addition, strong restacking of already-exfoliated monolayers is another cause of the low monolayer percentage, as verified by TEM CBED diffraction characterizations (Supplementary Fig. 6.8). However, by extending milling time, increasing rotation speed and improving rinsing, the percentage of monolayer h-BN was boosted to 57% (Supplementary Figs. 6.9, 6.10). The thin h-BN nanosheets show diffraction patterns following the reported finger-print of monolayer h-BN[9] under SAED technique (Fig. 4j–l) and typical single-layer height of ~0.6 nm under AFM characterization (Fig. 4m and Supplementary Fig. 6.11). Although the monolayer proportion in h-BN product is less than that of graphene, it still far exceeds those of

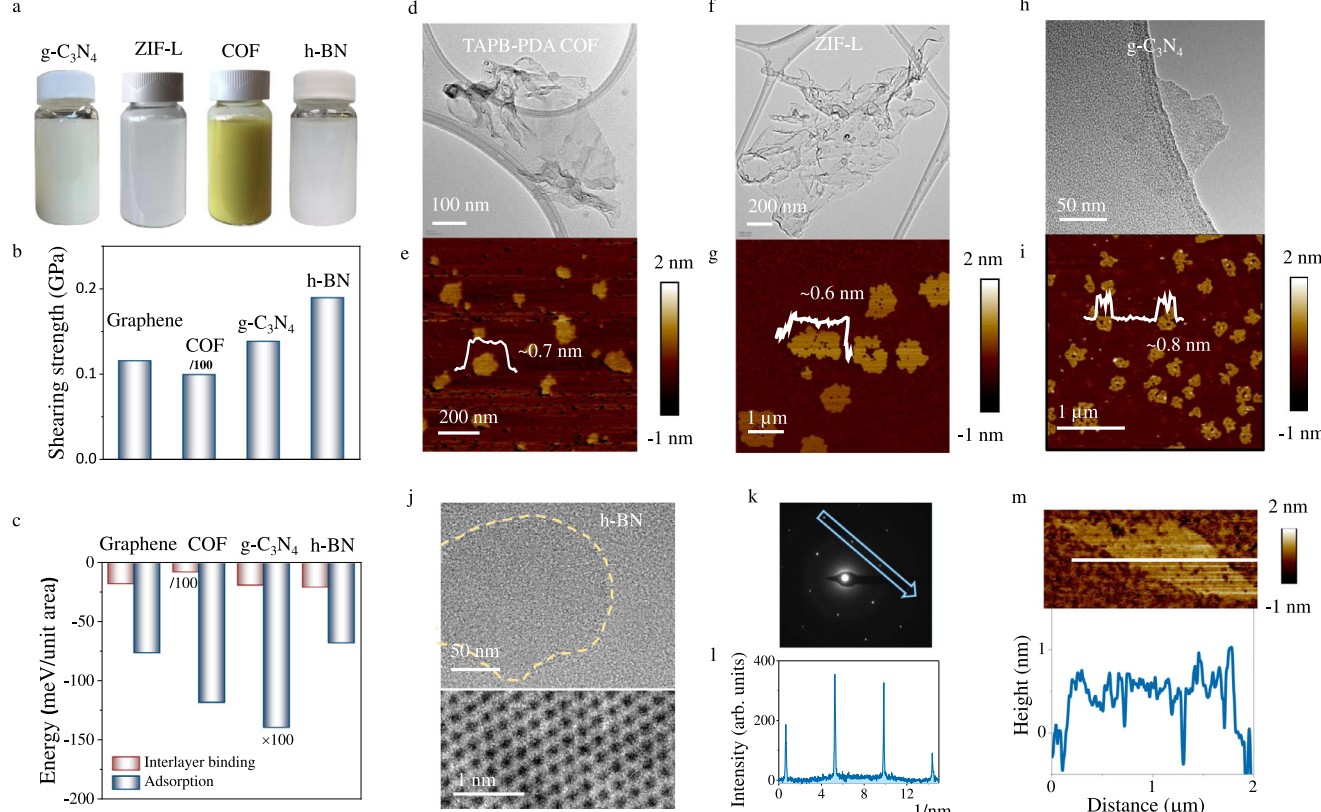

**Fig. 4 | Exfoliation of other layered materials by sticky milling. a** Photographs of exfoliated graphitic carbon nitride (g-C₃N₄), TAPB-PDA covalent organic framework (COF) and hexagonal boron nitride (h-BN) nanosheets dispersed in water at a concentration of 1 mg/mL. Zeolitic imidazolate framework (ZIF-L) nanosheets were dispersed in ethanol at a concentration of 1 mg/mL. **b** Comparison of the maximum interface shearing strength of graphene, TAPB-PDA COF, g-C₃N₄, and h-BN along the preferential sliding direction based on DFT calculation. Noted that the value for COF is multiplied by 100 to make it readable in this figure. This figure is based on the data from Supplementary Table 5.2. ZIF-L nanosheet was not studied by DFT calculation due to its structural instability (Supplementary Section 6.1.2).

**c** Comparison of interlayer binding energy of studied materials (Graphene, TAPB-PDA COF, g-C₃N₄, and h-BN) and the adsorption energy of PEI on their surface. To make them readable, we multiplied the value of the binding energy of TAPB-PDA COF by 100 and divided the value of the adsorption energy of PEI/ g-C₃N₄ by 100. This figure is based on the data from Supplementary Table 5.1. Selected TEM and AFM image of the exfoliated **d, e** TAPB-PDA COF, **f, g** ZIF-L, **h, i** g-C₃N₄. Insets are the height profiles of the nanosheets. **j** Selected low and atomic-resolution TEM images of the exfoliated BN nanosheets. **k** Selected area electron diffraction (SAED) pattern of a monolayer h-BN. **l** The intensity scan of the spots along the arrow in (**k**). **m** Selected AFM image and height profile of a single-layer h-BN.

currently adopted mechanical exfoliation strategies like ultrasonication, urea or sugar-assisted ball milling (<10%) (Supplementary Table 6.2)[15,17,38]. With proper adjustments implemented, it is reasonable to believe that this method can be tailored to exfoliate other layered materials with similar 2D configurations.

In summary, a scalable facile exfoliation method was present for exfoliating bulk layered materials with different structures into high quality monolayer nanosheets at high yield. Considering ball milling is a widely used technique in industry, this exfoliation technique arguably holds foreseeable potential to kilogram and even ton-scale by applying large milling jars and massive parallelization.

## Methods
### Viscosity control
In this sticky milling method, high-molecular weight branched Polyethyleneimine (PEI) with average Mn at ~10,000 by GPC, average Mw at ~25,000 by Light scattering (LS) (coded as high-viscosity PEI, h-PEI) (Sigma-Aldrich, 408727) was applied as a modifier to assist exfoliation. In order to assess the crucial role of high viscosity, low-molecular weight branched PEI with average Mw at ~800 by LS (coded as low-viscosity PEI, l-PEI) (Sigma-Aldrich, 408719) was chosen to mix with h-PEI at various ratios, giving PEI modifiers with viscosity at different levels, covering the range from 1508 to 150,882 mPa·s measured at ~20 °C (Supplementary Table 1.1).

### Sticky mechanical exfoliation
ZrO₂ milling jar and ZrO₂ balls with different weights and diameters (100 g ø d = 10 mm, 200 g ø d = 5 mm, and 20 g ø d = 0.1 mm) were used in this work. In a typical exfoliation process, pristine layered crystals (0.5 g) and 2 g PEI (high-viscosity PEI was used in most experiments unless stated) were loaded into 250 mL milling jars. The milling jars was loaded in a planetary ball mill (ZQM-P2, Changsha Mitr Instrument Equipment Co., Ltd) with its revolution radius at 10 cm and rotation radius equal to the radius of the milling jar at 39 mm. The rotation speed was set as 250 rpm for revolution and 500 rpm for rotation. Other parameters are listed in Supplementary Table 1.2. The whole process was conducted in ambient environment. After the milling process, 100 g milli-Q water was added to the milling jars. Then the milling jar was loaded into the mill again for 30 min at a lower speed (revolution: 150 rpm, rotation: 300 rpm) to disperse the nanosheets in water. Low speed was chosen in order to avoid liquid exfoliation during the dispersion process. The obtained nanosheets solution was repeatedly washed to remove excess PEI (Supplementary Section 1.5) and centrifugated at RCF of 236 g (Rotor 12181, Sigma 2-16 P) for 20 min to remove thick flakes.

### Other information
The rinsing process of graphene/PEI mixtures is described in Supplementary Section 1.5. The preparation methods for bulk g-C₃N₄, ZIF-L

and TAPB-PDA COF can be found in Supplementary Section 1.3. Characterization methods are provided in Supplementary Section 1.7, and more details, including sample preparation and parameter setting can be found in each of the characterization result discussions in Supplementary Information. The fabrication of flexible graphene conductive films and membranes with laminar ion channels, and the testing of ion permeation are provided in Supplementary Section 1.8. Details about DEM simulation and density functional theory (DFT) calculation are listed in the Supplementary Sections 5.2 and 5.5, respectively.

## Data availability

Relevant data supporting the key findings of this study are available within the article, the Supplementary Information and Source Data files. All raw data generated during the current study are available from the corresponding authors upon request. Source data are provided with this paper.

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

## Acknowledgements

We thank Yu Chen for helping with TEM characterization, Qiaoqiao Zhou for TGA analysis, Yingyi Huang for sheet resistance measurement, Jian Hu for zeta potential testing, Chenglong Xu for helping with AFM characterization, Anthony De Girolamo for performing elemental analysis, Yang Li for helping with SEM imaging, Xin Zeng for drawing graphics, Lilian Khaw and Loy Adrain for writing support. We acknowledge financial support from the Australia Research Council Research Hub under ARC Industry Transformation Research Hub for Energy-efficient Separation (IH170100009). X.Z. thanks the Australian Research Council for his ARC Future Fellowship (FT210100593). Q.H. thanks the Australian Research Council for his ARC DECRA Fellowship (DE180100266). This work made use of the facilities at the Monash Center for Electron Microscopy (MCEM) and Melbourne Center for Nanofabrication (MCN). DFT calculations were undertaken with the assistance of resources and services from the National Computational Infrastructure (NCI), which is supported by the Australian Government.

## Author contributions

Conceptualization was done by Z.W., X.Z., and H.W. The methodology was developed by Z.W. and X.Z. Characterizations were carried out by Z.W., Y.L., X Z., W.Z., X.L., S.Y., M.U., R.W., and Yan K. Computational study and simulations were designed and performed by X.Y., Q.H., and J.L. Mechanism discussion and study were done by S.L., Z.W., X.Z., and R.Q. Date analysis and validation were carried out by Z.W., Yuan K., Y.X., M.J., and L.G. The manuscript was written by Z.W. and revised by X.Z. and Yuan K. with contributions from all authors. All authors have read and agreed to the published version of the paper.

## Competing interests

The authors declare no competing interests.
