## [Peer Review File · Nature Communications]

Scalable High Yield Exfoliation for Monolayer NanosheetsREVIEWER COMMENTS

Reviewer #1 (Remarks to the Author):

The authors claim to have developed a method to produce single layer graphene by a polymer supported ball milling approach at high yields of the exfoliated product and monolayer contents of about 90%. According to the manuscript, the material can be dispersed in different solvents, resulting in colloidally stable nanosheet dispersions and polymer-free products. Acid-assisted defunctionalisation and calcination of deposited material at 600°C. The authors further show that following the procedures described in their manuscript, lamellar graphene films featuring high conductivity and charge selective ion permeation. Moreover, a mechanism for the exfoliation and the role of the PEI additive is discussed, supported by theoretic considerations. At last, the presented methodology is transferred to other 2D material systems for the production of nanosheet dispersions.

The following major points should be addressed by the authors in a revised manuscript before publication should be considered:

1.

The authors claim to yield ~90% powdered graphene nanosheets with an average size of 780 nm after 10 hours of milling time and >90% monolayers. However, this is not reflected in the XRD spectrum shown in Figure 1 b.

The authors should determine the change of the relative intensities for reflections with {001} contribution with respect to reflections with {100} or {010} character.

Additionally, measuring the relative intensities of powdered samples after different milling times can be used as indicator to confirm that the layer number is indeed decreasing for prolonged milling times.

Note that such changes can be illustrated by showing normalized spectra with horizontal offset.

2.

A major point are the AFM images: The deposits have a funny shape and homogeneous thickness. In typical exfoliation experiments, steps and terraces are observed from incompletely exfoliated or folded/wrinkled nanosheets. However, this is not observed in the AFM images shown in figure 1 or in figure 4. While the "incompletely exfoliated" nanosheet shown in Figure 3f shows terraces and steps, the observed step height is 0.3 nm and the shape of the nanosheet follows the shape of the terrace, which implies that the tip used for this measurement is worn. The features shown in figure 3f can therefore be considered as tip artefacts rather than incompletely exfoliated nanomaterial. This is further supported by the step height of 0.3 nm, which is three times lower than the typical graphene step height for polymer or surfactant assisted exfoliation products.

Note that, even minor contaminations in the solvent can dry in a funny way upon deposition, in particular the deposits in Figure S4.1 do not have characteristic shapes of nanosheets. Overall, additional experiments are required to confirm that the AFM deposits are indeed graphene. This is important since the high length/thickness aspect ratio of the nanosheets is a key point.

- To clarify if the observed features in the AFM are indeed nanosheets, the authors should relocate an area previously measured with AFM in the Raman microscope and map the same area. Typically, when depositing on Si/SiO₂, an interference contrast is observed that allows for relocalisation using the optical images (see examples here ACS Nano 2013, 7 (11), 10344-10353, Nano Lett. 2007, 7 (9), 2707-2710.). Optical micrographs should therefore also be provided to confirm that the deposits are homogeneous in thickness which should result in a similar colour. When using state of the art deposition techniques (e.g. ACS Applied Nano Materials 2020, 3 (12), 12095-12105), several metrics from Raman spectroscopy exist to identify the graphene thickness (Carbon 2020, 161, 181-189). Note that an analysis of the 2D band is not the most suitable, since decoupling of the layers through folding or intercalated solvent can decrease the apparent thickness when analysing the 2D band shape.

- Additionally, the authors could provide low-resolution wide view TEM images to confirm that the crystalline material observed in TEM has a similar shape as the objects in AFM (i.e. extended sheets with central holes) and not the typically observed

nanosheet shape with folds, wrinkles, terraces and straight edges

- If the authors can confirm that the flat objects in AFM are indeed the nanosheets, they should add a discussion to the main manuscript why they observe this different shape (e.g. "flowers" with central holes) in context with the current liquid exfoliation literature.

- The authors should give additional details on how the samples were deposited. And if / how the wafers were washed after deposition.

3.

The authors discuss a charge selective Ion permeation of produced lamellar graphene membranes. It should be clarified in the manuscript that the PEI was not removed by calcination for the ion diffusion experiments. While this is an important result, additional experiments should be considered. While PEI free graphene would be expected to show size selective ion permeation rather than charge selectivity. The role of the polymer during ion separation should be testified in a comparison.

4. The authors show the AFM size and thickness distribution of nanosheets after 1 and 2 hours of PEI assisted ball milling in Figure 3 a+b. It is evident from Figure 3a that the bulk material is exfoliated into small and thick, as well as large and thin nanosheets and the distribution approaches smaller and thinner nanosheet distributions after increasing the milling time, as shown in Figure 3b. However, this does not agree with the proposed mechanism as initially thicker nanosheets with a comparable lateral size distribution would be expected with a decreasing population of thicker and large nanosheets. This may be an issue with the number of counted nanosheets, which is also implied by the gap between the datapoints representing "large and thick" and "small and thin" nanosheets in Figure 3a. Typically no such splitting is observed in the data cloud unless solvent residues are mistaken for nanomaterial. However, it is not possible to identify the exact origin of this observation for the single nanosheets shown in Figure 3a+b. The authors should show an overview image with representative nanosheets for both milling times rather than examples of single sheets. Additionally, more nanosheet counts should be added for a representative discussion of the dimensions of exfoliated product.

Minor comments:

Line 15 and 50, hexagon boron nitride should be changed to hexagonal boron nitride
Figure 1c, Roman shift should be changed to Raman shift

Reviewer #2 (Remarks to the Author):

The manuscript presents the results of exfoliating graphene from graphite via ball-milling, using a viscous polymer as a solvent. While ball-milling is now a well-established method for production of graphene and related materials, the authors claim a very high yield, and an extraordinary yield of monolayer flakes. This claim is primarily supported by extensive AFM characterisation, and if true would be a significant advance. However I have concerns about the characterisation.

1. As well as AFM measurements, the authors show (limited) SEM images of flakes that had been filtered through alumina membranes. These show flakes with sharp edges, as is typically seen in top-down produced nanosheets. However, the AFM images tend to show objects with more rounded or serrated edges. I am curious about the discrepancy between the morphology seen in the two methods.

2. Before preparing the sample for AFM imaging, it was sonicated for 1 hr. The authors need to take care that this sonication is not carrying out further exfoliation. Also, for most accurate height measurements, profiles should be acquired along the fast-scan axis, to minimise effects of image flattening. See for example ISO TS 21356-1 Nanotechnologies – Structural characterization of graphene – Part 1: Graphene from powders and dispersions.

3. The objects measured and claimed to be monolayer are more than the 0.34 nm that would be expected from monolayer graphene flakes. While the presence of solvent/surfactant can lead to additional measured thicknesses, given the claim of such a high yield, it is important that there is some additional confirmation that these objects are indeed monolayer graphene. I would suggest Raman spectroscopy on an object that

has been measured by AFM, ideally more than one such object. See for example Paton et al. Nature Nanotechnology 13(6) DOI: 10.1038/nmat3944.

4. Measurement of total yield of material has been achieved gravimetrically, by filtering and weighing the filtered solids. It would be interesting to know how this compares to a value calculated by UV-vis extinction spectroscopy. For the yield claimed, there would have been very little material sedimented during the centrifugation step – was this the case, and could that also be quantified?

5. The other evidence of monolayer production is convergent-beam electron diffraction (CBED). The references for this refer to work that had used selected area electron diffraction (SAED) instead. I suggest the authors refer to Latychevskaia et al PNAS 115 (29) DOI: 10.1073/pnas.1722523115. Further details on the measurement is also needed, such as spot-size and convergence angle, as well as an indication on the bright-field image of where the diffraction is acquired from. The two bright-field images in fig S3-3 appear very different, with slightly different magnification, and possibly a different aperture used? Can the authors clarify the reasons for the differences?

6. For the XPS analysis, please show the full survey scans as well as the nitrogen peak. Can the authors also please explain the peak shifts seen in the nitrogen peak in fig. 2d? Is this due to charging or a change in chemical environment? Also provide information on the baseline function used to obtain the quantification. Does the nitrogen concentration from XPS agree with that from elemental analysis?

7. Overall, the presentation of the work could be clearer. In particular that specific steps taken to prepare samples for each characterisation method. It is currently difficult to follow in the extensive supporting information and can be easily clarified to make the paper much easier to read.

Reviewer #3 (Remarks to the Author):

The paper on Scalable High Yield Exfoliation of Monolayer Nanosheets is a thorough study of polymer protected ball-milling of layered materials. While there are previous reports on the matter, the authors designed a scalable process to achieve a high yield of monolayers.

From my perspective the conclusions should be taken carefully since for some reason the geometry of the exfoliated crystals seem by SEM (Fig. S3.1) is remarkably different with what was used for the AFM statistics. I worry in particular by figure S4.1 where it is clear they are basing statistics in what appears to be solvent or polymer drops. With this in mind I concur that some level of exfoliation was achieved from the Raman data, however I would not call it monolayer judging by the I2D/IG intensity ratio. I also base my conclusion on figure S3.3 that is clearly a bilayer.

Furthermore, I fail to see the reasoning in choosing the other layered materials tested since some of them were prepared in house. I would not call the exfoliation of carbon nitride successful as it is full of defects and, comparing with boron nitride, the problem could have come from the starting material.

Response Letter

Scalable High Yield Exfoliation of Monolayer Nanosheets

(NCOMMS-22-13685-T)

We greatly appreciate the valuable comments from editor and reviewers, and have conducted additional experiments to address these questions, particularly those concerning the characterization of graphene products. All **editorial** requirements in terms of publication policies, formatting, source data have been fulfilled as requested. The **reviewers'** comments and questions are responded point to point in this letter based on the new experiments and characterizations. Please be noted that the changes in the revised manuscript and supplementary information have been marked in red and heightened with black box in this letter.

Reviewer #1 (Remarks to the Author):

General comments: The authors claim to have developed a method to produce single layer graphene by a polymer supported ball milling approach at high yields of the exfoliated product and monolayer contents of about 90%. According to the manuscript, the material can be dispersed in different solvents, resulting in colloidally stable nanosheet dispersions and polymer-free products acid-assisted defunctionalisation and calcination of deposited material at 600°C. The authors further show that following the procedures described in their manuscript, lamellar graphene films featuring high conductivity and charge selective ion permeation. Moreover, a mechanism for the exfoliation and the role of the PEI additive is discussed, supported by theoretic considerations. At last, the presented methodology is transferred to other 2D material systems for the production of nanosheet dispersions.

Authors: We thank the reviewer for acknowledging the key breakthroughs made in this study and pointing out possible aspects where the work can be improved.

Specific comments: The following major points should be addressed by the authors in a revised manuscript before publication should be considered:

Q1. The authors claim to yield ~90% powdered graphene nanosheets with an average size of 780 nm after 10 hours of milling time and >90% monolayers. However, this is not reflected in the XRD spectrum shown in Figure 1 b. The authors should determine the change of the relative intensities for reflections with {001} contribution with respect to reflections with {100} or

{010} character. Additionally, measuring the relative intensities of powdered samples after different milling times can be used as indicator to confirm that the layer number is indeed decreasing for prolonged milling times. Note that such changes can be illustrated by showing normalized spectra with horizontal offset.

Authors: We agree with the reviewer’s point of referencing XRD spectra to reaffirm graphite delamination in ball milling process. To this end, we compared the XRD patterns of freeze-dried graphene powder obtained after 5h, 10h and 15h milling against that of graphite. Considering that (001) is a peak intrinsic to graphene oxide (GO) but not graphene¹, we instead studied graphene’s (002) peak, a characteristic peak generated by parallel layers as well, and examined its intensity relative to (100) peak. As shown in Figure R1.1, the relative intensities of (002) to (100) of these products after different milling times obviously weakened with prolonged milling time, thus implying gradually decreased layer number of the products with the process going on. It is though worth mentioning that a marginal (002) peak still remained for 15-h milled product, indicating that that few layers existed in the powder samples. We think that it is mostly because some dispersed monolayered nanosheets are prone to restacking in the freeze-drying process for XRD sample preparation. This hypothesis is further supported by the fact that increasing PEI (from 6.3 to 12.4 wt%, Supplementary Section S4.1) were able to reduce restacking during drying, leading to a weaker peak (002).

Powder samples	Intensity ratio of peak (002)/(100)
Graphite	56.65
5h-milling graphene	39.98
10h-milling graphene	18.66
15h-milling graphene	11.98

Response letter Fig. R1.1 The relative intensity of peak (002) to peak (100) of the graphene samples obtained from different milling time. Note that the reference intensity of peak (100) is obtained from the peak intensity between 40° to 50° of these powder samples.

Accordingly, the manuscript has been revised as follows in Figure 1 (new Figure 1b) and on page 3 of the main text:

Fig. 1. The physical properties of the graphene nanosheets via sticky mechanical exfoliation. **a**, Photograph of the obtained 110-g graphene in powder form. **b**, Powder XRD patterns of pristine graphite and graphene products after 5h, 10h, and 15h milling. **c**, Accumulated Raman spectrum of the graphene nanosheets after 15-h milling deposited on mica. Detailed characterization method can be found in supplementary S2.4. **d**, Selected AFM height profiles of graphene exfoliated by 5h, 10h and 15h milling, respectively (from left to right). **e**, HR-TEM image of exfoliated graphene (inset, atomic resolution). **f**, Converged beam electric diffraction (CBED) of graphene indicated by the yellow circle in e. Spot size: 9, convergence angle: ~15 milli-radians. **g**, Diffraction spot intensity taken along the lines in f. **h**, Comparison of the obtained graphene in this work with other mechanical exfoliated graphene in terms of lateral size and layer number (Detailed data in Supplementary Table S2.1). Blue dots: Liquid phase exfoliation; Orange dots: Solid phase exfoliation; Red dots: this work.

The crystalline structure of graphite is well preserved after this mechanical exfoliation, suggested by powder X-ray diffraction patterns (p-XRD) (Fig. 1b)². The graphene powder samples (freeze-dried from rinsed water solution) show broad peaks (100)/(101) at 43-45° and (110) at ~76°, corresponding to the 2D in-plane symmetry along graphene sheet. The decreased intensity of these peaks indicates the decreased lateral size with prolonging the milling time. All powder samples show a weak peak (002) at ~26°, a characteristic of paralleled graphene layers, which could mainly arise from nanosheet re-stacking during sample drying process. (Main text, Page 3)

Q2. A major point are the AFM images: The deposits have a funny shape and homogeneous thickness. In typical exfoliation experiments, steps and terraces are observed from incompletely exfoliated or folded/wrinkled nanosheets. However, this is not observed in the AFM images shown in figure 1 or in figure 4. While the “incompletely exfoliated” nanosheet shown in Figure 3f shows terraces and steps, the observed step height is 0.3 nm and the shape of the nanosheet follows the shape of the terrace, which implies that the tip used for this measurement is worn. The features shown in figure 3f can therefore be considered as tip artefacts rather than incompletely exfoliated nanomaterial. This is further supported by the step height of 0.3 nm, which is three times lower than the typical graphene step height for polymer or surfactant assisted exfoliation products. Note that, even minor contaminations in the solvent can dry in a funny way upon deposition, in particular the deposits in Figure S4.1 do not have characteristic shapes of nanosheets. Overall, additional experiments are required to confirm that the AFM deposits are indeed graphene. This is important since the high length/thickness aspect ratio of the nanosheets is a key point.

Q2A- To clarify if the observed features in the AFM are indeed nanosheets, the authors should relocate an area previously measured with AFM in the Raman microscope and map the same area. Typically, when depositing on Si/SiO₂, an interference contrast is observed that allows for relocalisation using the optical images (see examples here ACS Nano 2013, 7 (11), 10344-10353, Nano Lett. 2007, 7 (9), 2707-2710.). Optical micrographs should therefore also be provided to confirm that the deposits are homogeneous in thickness which should result in a similar color. When using state of the art deposition techniques (e.g. ACS Applied Nano Materials 2020, 3 (12), 12095-12105), several metrics from Raman spectroscopy exist to identify the graphene thickness (Carbon 2020, 161, 181-189). Note that an analysis of the 2D band is not the most suitable, since decoupling of the layers through folding or intercalated solvent can decrease the apparent thickness when analyzing the 2D band shape.

Authors: As per the reviewer’s suggestion, we conducted two additional characterizations to further confirm whether the flat objects observed from AFM are monolayer graphene, including (1) coupling AFM imaging with Raman spectroscopy and (2) AFM imaging of control samples with PEI alone.

(1) Coupling AFM imaging with Raman spectroscopy. Following the references^{3, 4}, We attempted to examine the graphene obtained after 15-h milling by correlating AFM topographic

imaging with Raman structure mapping. However, the original methodologies designed for these previous studies (ACS Nano 2013, 7 (11), 10344-10353, Nano Lett. 2007, 7 (9), 2707-2710.) were found not applicable in our case due to the drastic lateral size difference of graphene products. In the references, the physicochemical properties of graphene were determined by conducting AFM and Raman testing on exactly the same piece of nanosheet. This is achievable because the graphene extends for tens of microns and several nm in thickness, which makes its re-positioning fairly easy and precise under optical microscope during instrument shift. By contrast, our graphene product normally possesses lateral sizes less than one micron and thickness less than 1 nm, below the resolution limit of the optical microscope and thus making direct sample identification nearly impossible (Fig. R1.2). To address the problem, we alternatively employed a “mark-assisted strategy” to enable the AFM-Raman correlation: Firstly, 9 points are marked on a sample-loaded mica for visual guidance (Fig. S2.6A inset); Secondly, AFM probing is performed around these points for initial nanosheet spotting; Once spotted, nanosheets around this point, or marker, will subsequently be characterized by Raman spectroscopy. Accumulated Raman signals around these points indicate that there are many pieces of graphene in these areas as the obtained spectra showing identifiable typical D, G, 2D Raman peaks of graphene.

According to the recommended reference (Carbon 2020, 161, 181-189)⁵, G or 2D-protocols can be used for analyzing the layer number of the graphene nanosheets. Based on the described 2D-protocol, monolayer graphene should feature **a sharp and symmetrical 2D band**. This agrees with our case as the 2D band in our Raman spectra exhibits a symmetrical shape without identifiable shoulders. The G-protocol judges the Raman intensity of the substrate relative to the G band of graphene to identify the graphene thickness, which is based on the fact that the Raman signal of the substrate is affected by graphene thickness. However, we found the Raman intensity of the substrate in our case is always much higher than the signals coming from graphene (new supplementary Fig. S2.6). We reckon the reason lies in that the lateral size of the graphene in our case is much smaller than the Raman spot size. Since the substrate surface is not fully covered by graphene, part of the generated signals come from the uncovered substrate directly. In addition, G-protocol needs a piece of monolayer graphene as the benchmark for calculating the layer number of others, which is also proved to be challenging in this case. Alternatively, we studied the intensity of G band relative to the intensity of 2D band to identify the thickness of our graphene nanosheets. **The intensity ratios of I(2D)/I(G) of these generated Raman patterns locate in the range of 1.06-3.1** (Fig. S2.6C). Given that

the $I(2D)/I(G)$ band ratio is lower than 1 for bilayer graphene, and the ratio decreases with the increase of layer number⁵. Therefore, we conclude that the sub-nm discs observed by AFM are mostly monolayer graphene.

Response letter Fig. R1.2. Photo of graphene deposited on mica obtained by optical microscope of Raman device.

Supplementary Fig. S2.6. point marker spotted by the optical microscope of AFM (A) and Raman device (B). Inset image in A is the photograph of the Raman sample with 9-point markers. (C) AFM images (TOP) and Raman spectra (Bottom) obtained from the points indicated in A.

We have added the details of the new experiments in the supplementary and result discussion was also added as follows.

S2.4 Further evidence on monolayer graphene

To provide further evidence for validating that the small discs spotted by AFM are graphene nanosheets, we tried to correlate the AFM morphology imaging with Raman structure mapping. Graphene fabricated from 15-h milling was deposited on mica following the same procedures of AFM sample preparation and used for demonstration. Since our nanosheets were not easily observed by Raman and AFM optical microscope, we manually marked our sample with 9 points to help us locate the same area in two separated characterization devices. The AFM characterization was firstly performed to find nanosheets around these points. As shown in the top row of Figure S2.6C, small discs were found in the area 1, 2, 3 with their thickness at around 1 nm and lateral size around 200 nm similar to what we found previously from AFM (Fig. 2.6A). Once nanosheets were spotted in the vicinity of one of these points, Raman spectroscopy was then utilized to generate spectra from the same area with the help of our point markers on mica (Fig. S2.6B). Repeatedly accumulated Raman signals indicate that there are plenty of pieces of graphene in these areas with identifiable typical D, G, 2D Raman peaks of graphene. All Raman patterns feature a symmetrical 2D band, suggesting the high monolayer percentage^{5, 6}. Besides, the intensity ratios of $I(2D)/I(G)$ of these generated Raman patterns locate in the range of 1.06-3.1 (Fig. S2.6C). Giving that the $I(2D)/I(G)$ band ratio is lower than 1 for bilayer graphene, and the ratio decreases with the increase of layer number⁵. In this line of reasoning, we conclude that the sub-nm discs observed by AFM are monolayer graphene.

Besides Raman spectroscopy, these small graphene discs were further observed by low resolution TEM and their diffraction patterns were obtained at the same time. The observed graphene nanosheets show a weak contrast to the carbon mesh background with their lateral size locates around 200nm, which is line with the result of AFM (Fig. S2.7). Two types of diffraction patterns of the obtained graphene nanosheets were generated, with converged beam electric diffraction (CBED) was applied on part of ultra-large graphene nanosheets (Fig. 1e-g, main text) and selected area electron diffraction (SAED) was performed on the whole piece of small ones (Fig. S2.7). These patterns exhibit a typical six-fold symmetry as expected from graphene or graphite. Besides, intensity scans along these diffraction patterns more intense

inner spots ((0-110) and (-1010)) than outer ones ((1-210) and (-2110)), with an intensity ratio ($I_{\text{inner spots}}/I_{\text{outer spots}} > 1$), which is a characteristic fingerprint of monolayer graphene⁶.
(Supplementary, Page 17-18)

In addition, a new AFM image of incompletely exfoliated graphene nanosheet with uniform steps has been used to replace Fig 3f to avoid misleading readers.

(2) AFM imaging of pure PEI as control experiment was also conducted to check whether PEI molecules can form 2D shapes after drying. Diluted PEI solutions with different concentrations were deposited on mica for AFM characterization using the same preparation procedure of graphene samples. Given that the residual PEI in the graphene samples was in the range of 0.315-0.62 $\mu\text{g/mL}$ (calculated based on CHNS elemental analysis), three PEI concentrations (1, 0.5 and 0.05 $\mu\text{g/mL}$) were chosen. As shown in Fig. R1.3, PEI could not form any types of regular shapes with identifiable z-direction steps at all the three samples. Only highly rough surface with ripples perpendicular to the scan axis of the cantilever was observed, most likely resulting from strong drag effect by polymer chains of PEI. This drag effect-caused roughness of PEI on AFM characterization was also observed previously on PEI functionalized graphene oxide⁷. The results exclude the possibility that the 2D sheets observed under AFM were PEI contaminants.

Response letter Fig. R1.3 AFM characterization of three samples with PEI alone at difference concentrations on mica.

Q2B- Additionally, the authors could provide low-resolution wide view TEM images to confirm that the crystalline material observed in TEM has a similar shape as the objects in AFM (i.e. extended sheets with central holes) and not the typically observed nanosheet shape with folds, wrinkles, terraces and straight edges. If the authors can confirm that the flat objects in AFM are indeed the nanosheets, they should add a discussion to the main manuscript why they observe this different shape (e.g. “flowers” with central holes) in context with the current liquid exfoliation literature.

Authors: As suggested, low-resolution TEM imaging coupled with the SAED diffraction technique was conducted. As shown in Fig. S2.7, the observed graphene nanosheets displayed similar shapes as observed by AFM and with their lateral size locates around 200nm, despite a relatively weak contrast to the carbon mesh background. The diffraction patterns of these nanosheets exhibit a six-fold symmetry and characteristic fingerprint of monolayer graphene with more intense inner spots than outer ones.

We agree with the reviewer that the structure of large nanosheets in Fig. S4.1 seems unusual. To understand why **“flower-like” structure** was formed, we conducted AFM imaging under high resolution using low scan rate and realized that the flower structure was actually formed due to the aggregation of many small discs (Fig. R1.4A). Considering its sub-1-nm height and flat shape, we believe these small discs are graphene nanosheets. To elucidate the origin of their aggregation, we reviewed the preparation procedure of our AFM samples. In our study, we prepared the AFM samples by loading ethanol-based sample solution onto freshly cleaved mica so that the dispersed nanosheets would settle during solvent evaporation. While this process was supposed to occur simultaneously across the mica surface, the solvent evaporation rate exhibited local variations across the sample. This led to the transition of the initial ethanol layer into many ethanol droplets surrounded by dry areas as shown in Fig R1.5. While nanosheets in dry areas were spread to show even and discrete distribution, we believe that those in the remaining droplets became concentrated with ongoing evaporation and deposited collectively at last, showing a flower-like structure.

Moreover, we found that the feature to distinguish the “flower-like” nanosheet assembly from a real single large nanosheet is the hollow structure. Under high-resolution AFM imaging mode, the single large nanosheets showed continuous surface (Fig. R1.4 B). Please be noted that the unusual “flower-like” sheets were treated as PEI residues in our previous manuscript and were not counted in statistical lateral size analysis.

Response letter Fig. R1.4. (A) High-resolution AFM imaging on ultra-large “flower-like” structure existed in previous graphene AFM samples. (B) High-resolution AFM imaging on large graphene nanosheets.

Response letter Fig. R1.5. Photographs of graphene solution evaporation process on mica.

As we realized that such “flower-like” structures could cause confusion and misunderstanding, we have provided new AFM images (Supplementary Fig. S2.3) and revised discussion to avoid confusion.

Corresponding revision in revised main text.

When rinsed graphene solution was deposited on mica, plenty of small flat discs with a lateral size of hundreds of nanometres and a height between 0.5 to 1 nm were observed (Fig. 1d). In addition, their apparent heights exhibit a narrow unimodal distribution with deviations less than 0.2 nm (Fig. 1d, Supplementary Figs. S2.2, S2.3). These AFM characteristics suggest that these observed sub-nm discs may be monolayer graphene^{22, 23}. To confirm the finding, we collected accumulated Raman signals from these discs on mica (Supplementary Fig. S2.6). The generated patterns feature characteristic D, G, and 2D bands of graphene. The intensity ratios of $I(2D)/2(G)$ locate in the range of 1.06 to 3.1, implying high monolayer percentage in the spotted graphene⁸ (Fig. 1c and Supplementary Fig. S2.6). In addition, the intensity ratio of D/G band, which is widely used as an indicator of sp^3 carbon defects, is between 0.38 and 0.44 for

our graphene samples. The D/G ratio is lower than those of graphene oxide (GO) and reduced GO (rGO) ($\sim 0.7-0.9$)²⁴, indicating the much preserved high-quality lattice of the as-prepared graphene nanosheets. Transmission electron microscopy (TEM) characterization also shows that these graphene nanosheets have lateral sizes of hundreds of nanometers, in agreement with AFM characterization. The weak contrast of these nanosheets to TEM carbon grid reveals that they are ultrathin. As illustrated in Figs. 1e-g and Fig. S2.7, the TEM diffraction patterns display a typical six-fold symmetry and characteristic fingerprint of monolayer graphene with more intense inner spots ($\{1010\}$ facets) than outer ones ($\{2110\}$ facets)^{8, 10, 25}, providing further evidences that the as-exfoliated products are dominated by monolayer graphene. **(Main text, Page 4)**

As for the shape difference between graphene nanosheets obtained via typical liquid exfoliation methods (sharp edges) and that in this study (irregular serrated edges), we believe it to be caused by different nanosheet in-plane breaking mechanisms under these two circumstances.

In ultrasonication-assisted liquid exfoliation, material breaking is achieved in two steps comprising deformation and delamination. The ultrasound wave evenly propagating through the material will first generate numerous in-plane crystalline structural deformations such as basal plane slip¹³ and twining¹⁴, forming mechanically vulnerable “kind band striations” along zigzag directions on material surface¹⁵. Solvent intercalation and solvation will then assist delamination, thus producing nanosheets with sharp edges. The ball milling process, however, enables in-plane breaking primarily by the collision between materials and grinding balls. Our observation has revealed the rough surface of these balls featuring protruding sharp ridges (Fig. 3b), which we found providing sufficiently high compression to break graphene nanosheets at any points due to concentrated forces. Therefore, the shape of delaminated nanosheets is largely dependent on the relative position of the ridges involved in each in-plane breaking event. Considering the highly random distribution of these ridges over the balls, it is reasonable that as-obtained graphene nanosheets possess more irregular edges.

Noteworthy, the above explanation is further supported by our comparison of nanosheets produced by the two methods in previous studies. In these studies, nanosheets obtained by ball milling^{16, 17, 18, 19} overall tend to show a more irregular shapes with serrated edges than nanosheets obtained by sonication exfoliation methods^{6, 20, 21}. This implies that the shape

difference is intrinsic to nanosheet preparation method. Discussion on shape difference has been added in the revised version:

“Since the in-plane breaking is primarily because of the protruding tips on the surface of grinding balls, the randomly distributed tips lead to a random in-plane breaking in terms of directions and locations on nanosheets accordingly. This random breaking process differs from the typical LPE exfoliation processes in which breaking occurs along the crystalline structure deformation directions, mostly along zigzag direction. As a result, the obtained nanosheets (Fig. 1d) as well as other nanosheets from ball milling processes^{16, 17} tend to show more irregular edges (serrated) than that of nanosheets from sonication exfoliation strategies with sharp edges^{6, 21}.” (Main text, Page 8)

Q2C- The authors should give additional details on how the samples were deposited. And if / how the wafers were washed after deposition.

Authors: As requested, the AFM sample preparation procedures have been added in the revised manuscript in Section 1.7.

S1.7 Characterizations

The thickness and lateral size of the exfoliated nanosheets were characterized by atomic force microscopy (Bruker Dimension Icon). To prepare the samples, the rinsed water dispersion of graphene nanosheets was diluted with ethanol to a concentration of 5 $\mu\text{g/mL}$ after sufficiently washing out free PEI molecules. The graphene/ethanol solution was then sonicated for 1 h using a Unisonics FXP12M sonic bath (40 kHz, 100 W). Freshly exfoliated mica (12.5 mm, grade V-1, ProSicTech) plates were chosen as the substrates and the exposed new surface was washed with ethanol after exfoliation using scotch tape. Before the ethanol on the mica totally dries, 3 μL of the diluted dispersion was dropped onto it using a pipette, and then carefully shook the mica in order to ensure the ethanol dispersion is evenly distributed on the surface. The mica with graphene nanosheets on top was then left in a 45 $^{\circ}\text{C}$ vacuum oven overnight before AFM characterization. Tapping mode was chosen for the analysis, and a relatively hard cantilever (TESPA-V2, BRUKER) was applied accordingly. (Supplementary, Page 8-9)

Q3. The authors discuss a charge selective Ion permeation of produced lamellar graphene membranes. It should be clarified in the manuscript that the PEI was not removed by calcination for the ion diffusion experiments. While this is an important result, additional experiments should be considered. While PEI free graphene would be expected to show size selective ion permeation rather than charge selectivity. The role of the polymer during ion separation should be testified in a comparison.

Authors: Thank you for raising this question. To evaluate the role of PEI in regulating ion transport, we fabricated three types of laminar membranes using graphene nanosheets exfoliated after 10-h milling. The first one was made of as-prepared graphene on PES substrate. The second membrane was made of as-prepared graphene on PES substrate and treated with 0.1M HCl. The third membrane was made of as-prepared graphene nanosheet on AAO substrate and followed by a thermal treatment at 450 °C. Different substrates were used due to the poor thermal stability of PES and the poor acid resistance of AAO substrate. We found HCl-treated membranes show a decrease on both ion transport rate and divalent/monovalent ion selectivity (Fig. S4.4). This indicated the two main roles of PEI in the membrane^{7, 22, 23} of (1) acting as molecular spacers to increase the mass transport rate; and (2) regulating ion transport by the Donnan effect. However, the calcined membrane where PEI was totally removed displayed an unexpected ion diffusion rate that was two magnitudes higher than that of the pristine one, while showing no ion selectivity. This could be attributed to the pinholes formed during the incineration process, which is often seen in the synthesis of inorganic membranes.

Corresponding revision has been made in the revised manuscript:

To highlight its charge property and high aspect ratio, graphene nanosheets were parallelly piled into a membrane with laminar channels (Fig. 2f inset). The laminar membrane showed an exceptional fast transport of monovalent ions (Na^+ , K^+) at a rate up to $3.8 \text{ mol m}^{-2} \text{ h}^{-1}$ while a relatively low transport rate of divalent ions (Ca^{2+} , Mg^{2+}), leading to a competitive mono/divalent ion selectivity of 5.11~7.0 (Fig. 2f). **An apparent drop in mono/divalent selectivity was observed after the membrane was treated by HCl to partially remove PEI, demonstrating the role of PEI in regulating ion transport (Supplementary Fig. S4.4).** This ion transport behavior in positive-charge governed nanoconfinement could be leveraged for ion-separation, energy harvesting and storage, and sensing technologies^{7, 23, 24, 25, 26}. **(Main text, page 6)**

The new experiment results have also been added to supplementary information:

S4.5 Ion transport of HCl-treated membranes

We found membranes that have experienced HCl de-functionalization show a diffusion rate of monovalent ions (Na^+ , K^+) at $\sim 1.9 \text{ mol m}^{-2} \text{ h}^{-1}$ and divalent ions (Ca^{2+} , Mg^{2+}) of $\sim 0.5 \text{ mol m}^{-2} \text{ h}^{-1}$, leading to a moderated mono/divalent ion selectivity of ~ 2.9 to 4.4 (Fig. S4.4). The decreased ion transport rate and divalent/monovalent ion selectivity emphasize the main roles of PEI in the laminar membrane^{7, 22, 23}: (1) acting as molecular spacers to increase the mass transport rate; (2) regulating ion transport by the Donnan effect. (Supplementary, Page 30)

Supplementary Fig. S4.4. Ion permeation of HCl-treated membranes

Q4. The authors show the AFM size and thickness distribution of nanosheets after 1 and 2 hours of PEI assisted ball milling in Figure 3 a+b. It is evident from Figure 3a that the bulk material is exfoliated into small and thick, as well as large and thin nanosheets and the distribution approaches smaller and thinner nanosheet distributions after increasing the milling time, as shown in Figure 3b. However, this does not agree with the proposed mechanism as initially thicker nanosheets with a comparable lateral size distribution would be expected with a decreasing population of thicker and large nanosheets. This may be an issue with the number

of counted nanosheets, which is also implied by the gap between the datapoints representing “large and thick” and “small and thin” nanosheets in Figure 3a. Typically no such splitting is observed in the data cloud unless solvent residues are mistaken for nanomaterial. However, it is not possible to identify the exact origin of this observation for the single nanosheets shown in Figure 3a+b. The authors should show an overview image with representative nanosheets for both milling times rather than examples of single sheets. Additionally, more nanosheet counts should be added for a representative discussion of the dimensions of exfoliated product.

Authors: Thank you for the critical comments. In line with them, we first enlarged our counting scope to include more samples in the updated morphology distribution figures (Fig S5.1). We also presented the data using liner axis instead of the previously used logarithmic axis, which we believed to cause the unusual data splitting. The updated figures show that 1-h milled products display morphology in three regions, namely “thick-and-small”, “thin-and-large” and “thick-and-large” while 2-h milled products fall into only the first two regions. Unfortunately, even for 1-h milled products, the proportion of “thick-and-large” samples is still smaller than expected, which could be attributed to AFM sample preparation.

The AFM samples of 1-h and 2-h milling products were prepared in three steps of water rinsing (to remove residue PEI from milled mixture), ethanol dilution (to facilitate sample dispersion and drying on mica) and sample transferring/loading on mica. To preserve products with all possible morphologies for later observation, the sample was prepared without pre-centrifugation. While thin nanosheets and small thick few layers are stably dispersed in ethanol, large-and-thick unexfoliated particles could settle quickly due to gravity, making them unintentionally being missed out at the sample transferring/loading step.

Although the absolute number of “thick-and-large” samples is hardly obtainable due to imperfect sample preparation, the semi-quantitative comparison between 1-h and 2-h products still reveals the main transitional trend in the product morphology, that is nanosheets become thinner and smaller and gradually resulting in most “thin-and-small” nanosheets (Fig. R1.6). These still effectively support our proposed milling exfoliation mechanisms based on simultaneous breaking and delamination by grinding effect.

However, we fully agree with the reviewer that overview images with representative nanosheets would be much better at explaining the exfoliation progress in our ball-milling process than confusing statistical results. Therefore, the AFM images are added as the revised Fig. 3a and 3b while the updated statistical data are moved to the supplementary information.

The two new AFM images should have exhibited the contrast morphologies between 1-h and 2-h products. Corresponding discussions have also been revised in the sections below. It is, however, worth mentioning that these large-and-thick unexfoliated particles are removed by pre-centrifugation for any other testing, counting or characterizations in our study so they will not compromise other discussions and results.

Corresponding revisions of Fig 3.

Fig. 3 Exfoliation mechanism of the sticky exfoliation. a, AFM characterization on graphene products obtained after 1-h and 2-h milling, respectively.

Revised statistic data has been moved to SI:

Revised discussion in SI:

To understand the exfoliation mechanism, graphene obtained from 1-h and 2-h milling was deposited on mica by following the same AFM sample preparation procedures excepting without centrifugation. We found that both thin-yet-large and thick-yet-small nanosheets exist in the exfoliation products at 1-h and 2-h (Fig. S5.1). The absence of large and thick particles could be the result that they are prone to precipitation and thus being excluded during sample preparation process. Along with the milling time increasing to 2 h, the lateral size of the nanosheets narrows down to 1 to 2 μm , and their thickness reduces to less than 100 nm (Fig. S5.1, right).

According to these results, the proposed exfoliation and material breaking mechanism presents as follows. Starting from thick and large particles, the in-plane breaking and out-of-plane delamination occur simultaneously on them at the initial stage driven by grinding effects. The delamination process could start from the very top/bottom surface of these bulk particles, delivering ultra-thin nanosheets and leaving thick parent particles. With further grinding, chances of breaking will dramatically be reduced with the decreasing thickness or when the lateral size of the nanosheets is comparable to the surface roughness of the grinding balls (Fig. S5.6). As a result, the lateral size distribution narrows with the prolonging of milling time. The exfoliation is further evidenced by an incompletely exfoliated graphene shows height steps of ~ 0.7 nm (Supplementary Fig. S5.2) (**Supplementary, Page 31**)

Revised discussion in the main text:

AFM topographic images show that both thin-yet-large and thick-yet-small nanosheets exist in 1-h exfoliation product (Fig. 3a, Supplementary Fig. S5.1). The distributions of thickness and lateral size gradually narrow down with extending milling time to 2 h. The results indicate that graphite particles undergo breaking and delamination simultaneously during the milling process. **(Main text, page 7)**

Minor comments:

Line 15 and 50, hexagon boron nitride should be changed to hexagonal boron nitride
Figure 1c, Roman shift should be changed to Raman shift

Authors: The typo has been corrected.

Reviewer #2 (Remarks to the Author):

General comments: The manuscript presents the results of exfoliating graphene from graphite via ball-milling, using a viscous polymer as a solvent. While ball-milling is now a well-established method for production of graphene and related materials, the authors claim a very high yield, and an extraordinary yield of monolayer flakes. This claim is primarily supported by extensive AFM characterization, and if true would be a significant advance. However, I have concerns about the characterizations.

Authors: We appreciate that the reviewer acknowledges the potential impact of this study.

Specific comments: Q1. As well as AFM measurements, the authors show (limited) SEM images of flakes that had been filtered through alumina membranes. These show flakes with sharp edges, as is typically seen in top-down produced nanosheets. However, the AFM images tend to show objects with more rounded or serrated edges. I am curious about the discrepancy between the morphology seen in the two methods.

Authors: We thank the reviewer for pointing out this issue. To understand the origins of the discrepancy between AFM and SEM images, we did more experiments detailed as follows. We found that there was a **SEM observation bias** on large nanosheets caused by sample preparation and resolution limitation.

Sample preparation. For SEM characterization, the graphene dispersion in water was placed on AAO substrate. Considering the approximate size of AAO pores (100~300 nm) to that of our product, it is expectable that a certain fraction of nanosheets will pass through the pores and only those with larger lateral size have a better chance to stay for SEM characterization. Our supporting filtration experiment coupled with UV-Vis spectra corroborates this notion by showing that over 80% of nanosheets in the dispersion penetrated the AAO support (Fig. R2.1A). There was no such selection for AFM samples in which graphene nanosheets were deposited on non-porous mica substrate.

Resolution limitation. Even when small graphene nanosheets are successfully deposited, their monolayered nature endows them fairly low contrast against the support that severely compromises their visibility. One example as Fig. R2.1B illustrates the visual difference of a few nanosheets under SEM. While a larger piece of nanosheet in the middle (red cycle) displays clear shape, smaller nanosheets in the yellow circles are somewhat elusive. Upon close

observations, we notice that these small nanosheets possess round shapes with serrated edges just as what we have seen from AFM. By contrast, the large one with sharp edges is possibly the stack of several nanosheets with self-folding or multi-layers that have not been fully exfoliated.

Response letter Fig. R2.1 (A) Comparison of UV-Vis adsorption between SEM sample preparation solution before and after filtration through AAO membrane. (B) SEM image of graphene deposited on AAO substrate.

Low-resolution TEM coupled with the SAED diffraction technique as a reference. To verify our hypothesis on the observation bias from SEM, we used low-resolution TEM to interrogate nanosheet samples with serrated edges or sharp edges and compared their crystallinity under diffraction mode. It was found that graphene nanosheets spotted with serrated edges not only possess similar lateral dimension to the monolayered graphene seen in AFM, but also display a clear mono-crystalline diffraction pattern, implying its single-layer nature (Fig. S2.7). However, sharp-edged and large nanosheets that have been seen in SEM images showed multiple sets of diffraction spots, an indicative of either self-folded edges or multilayered stacks (Fig. R2.3).

Supplementary Fig. S2.7. Low resolution TEM images of the graphene nanosheets (15-h milling). Insets are the SAED diffraction patterns of the spotted graphene.

Response letter Fig. R2.3 TEM image and diffraction patterns of ultra-large graphene nanosheets spotted in 15-h milling sample.

The relationship between size and roundness. To further study the shape difference between large nanosheets and small ones, a statistical analysis on the shape of the graphene nanosheets based on intensive AFM imaging was conducted. We found a negative correlation between the lateral size of graphene nanosheets and their roundness (Fig R2.2). This negative correlation could be the result of the different in-plane breaking processes. Initial delamination and breaking may start from peeling or tearing along mechanical vulnerable crystalline structural deformations on material surface, often named “kind band striations”^{13, 14, 15}. This initial crystalline structural breaking will deliver incompletely exfoliated few-layer nanosheets with large sizes and sharp edges, a few of them are left to be observed. The majority of large nanosheets then undergo further in-plane breaking by collisions of protruding sharp ridges on milling ball surface (Fig. 3b), which we found to provide sufficiently high compression to break graphene nanosheets at any points due to concentrated forces. Considering the highly random distribution of these ridges over the balls, the long edges of the nanosheets have a better chance to be trimmed than short edges during milling, thus smaller nanosheets that have undergone more times of breaking tend to be more likely to end up with higher roundness but with serrated edges.

Response letter Fig. R2.2 Statistical analysis of the relationship between the lateral size of the exfoliated graphene nanosheets and their roundness measured by image J software.

In light of these results, we replaced the misleading SEM images with low-resolution TEM images to provide coherent morphology and size information to readers. It should be noted that the bias in our previous SEM observation does not affect the reliability of the results of the size and monolayer percentages as they were calculated based on AFM characterization. The manuscript has been revised as follows:

Besides Raman spectroscopy, these small graphene discs were further observed by low resolution TEM and their diffraction patterns were obtained at the same time. The observed graphene nanosheets show a weak contrast to the carbon mesh background with their lateral size locates around 200nm, which is line with the result of AFM (Fig. S2.7). Two types of diffraction patterns of the obtained graphene nanosheets were generated, with converged beam electric diffraction (CBED) was applied on part of ultra-large graphene nanosheets (Fig. 1e-g, main text) and selected area electron diffraction (SAED) was performed on the whole piece of small ones (Fig. S2.7). These patterns exhibit a typical six-fold symmetry as expected from graphene or graphite. Besides, intensity scans along these diffraction patterns more intense inner spots ((0-110) and (-1010)) than outer ones ((1-210) and (-2110)), with an intensity ratio ($I_{\text{inner spots}}/I_{\text{outer spots}} > 1$), which is a characteristic fingerprint of monolayer graphene⁶. It should be mentioned that agglomeration, overlapping and folding of graphene nanosheets are more commonly encountered than in the case of AFM characterization possibly as the result of different sample preparation strategies. **(Supplementary, Page 18)**

The graphene nanosheets were then investigated by transmission electron microscopy (TEM), exhibiting a weak contrast to the carbon mesh background with lateral size at a scale of hundreds of nanometers, in agreement with AFM characterization. The layer number information of these observed thin graphene nanosheets were gained through their diffraction patterns. As illustrated in Figs. 1e-g and supplementary Fig. 2.7, their diffraction patterns display a typical six-fold symmetry and characteristic fingerprint of monolayer graphene with more intense inner spots ($\{1010\}$ facets) than outer ones ($\{2110\}$ facets)^{6, 11, 12}. **(Main text, page 4)**

Q2. Before preparing the sample for AFM imaging, it was sonicated for 1 hr. The authors need to take care that this sonication is not carrying out further exfoliation. Also, for most accurate height measurements, profiles should be acquired along the fast-scan axis, to minimise effects of image flattening. See for example ISO TS 21356-1 Nanotechnologies — Structural characterization of graphene — Part 1: Graphene from powders and dispersions.

Authors: Thank the reviewer for the suggestion. In order to evaluate the impact of sonication for AFM sample preparation, we conducted additional experiments to compare three AFM samples with one subject to **1-h** milling with **3-h** sonication, one subject to **15-h** milling but **5-min** sonication, and one with **15-h** milling and **1-h** sonication. The thickness profile of the sample after short milling (1h) with intensive sonication (3h) shows a poor exfoliation efficiency as the majority of the spotted objects are still small thick flacks (Fig. R2.4A). By contrast, the sample after long milling (15h) with short sonication (5min) contains thin nanosheets (Fig. R2.4B). The results indicate the much more important role of milling itself in materials exfoliation than sonication.

Although the thickness of the obtained graphene (15-h milling and 5-min sonication) has been exfoliated to sub-1 nm scale, they tend to show a rougher surface (Fig. R2.4B). Previous report⁷ has pointed that PEI residues can lead to a high-roughness surface observed by AFM, which means sonication could help PEI molecules desorb from nanosheets so as to give a smoother surface under AFM characterization (Fig. R2.4C). Besides, we also noticed that sample without sonication tends to have a bit larger lateral size with an average of ~300 nm than that of 15h milling and 1-h sonication sample (Fig. R2.4 B and C).

Besides, height profiles of all AFM figures in manuscript and supplementary information have been changed to take along the fast-scan axis as suggested.

Response letter Fig. R2.4. Selected AFM image of graphene obtained by 1-h milling with 3-h sonication. (B) Selected AFM image of graphene obtained by 15-h milling and 5-min sonication. (C) Selected AFM image of graphene obtained by 15-h milling and 1-h sonication.

Q3. The objects measured and claimed to be monolayer are more than the 0.34 nm that would be expected from monolayer graphene flakes. While the presence of solvent/surfactant can lead to additional measured thicknesses, given the claim of such a high yield, it is important that there is some additional confirmation that these objects are indeed monolayer graphene. I would suggest Raman spectroscopy on an object that has been measured by AFM, ideally more than one such object. See for example Paton et al. Nature Nanotechnology 13(6) DOI: 10.1038/nmat3944.

Authors: Thank the reviewer for these suggestions, which happened to be raised by Reviewer 1 as well. We kindly invite the review to check on our additional experiments that should answer this question, including coupling Raman spectroscopy with AFM, low-magnification TEM, control experiments on pure PEI. Please refer to our response to Question 2 of Reviewer 1 (**Response letter, Page 4-9**).

Q4. Measurement of total yield of material has been achieved gravimetrically, by filtering and weighing the filtered solids. It would be interesting to know how this compares to a value calculated by UV-vis extinction spectroscopy. For the yield claimed, there would have been very little material sedimented during the centrifugation step – was this the case, and could that also be quantified?

Authors: We thank you for your valuable comment. UV-Vis spectroscopy of graphene (15h milling) water solutions at different concentrations have been provided. The UV-Vis spectra

are found to provide results as reliable as that of gravimetric calculation in terms of exfoliation yield. The detailed procedures and results have been added in the revised SI.

S2.1 Exfoliation yield

The exfoliation yield was mostly evaluated gravimetrically by weighting the solid content in the rinsed graphene solutions. For the sake of measuring solution concentration efficiently, UV-Vis spectroscopy of graphene (15h milling) water solution at different concentration is provided in Fig 2.1A. Like other mechanical exfoliation methods, graphene obtained here also displays a linear relationship between its concentration and adsorption in the visible range. We then used the adsorption of graphene solution at wavelength of 700 nm and correlated it with the concentration. A clear linear correlation was fitted and a mathematical equation was provided to evaluate the concentration by UV-Vis spectroscopy (Fig. 2.1B). The obtained equation was validated by calculating the exfoliation yield of other tanks that have experienced the same milling program, giving reliable results with deviations less than 5 %.

Supplementary Fig. S2.1. (A) UV-Vis spectroscopy of graphene (15h milling) water solution at different concentration. (B) Calibration standard curve of graphene nanosheet dispersions using their UV-Vis absorption at 700 nm. (**Supplementary, Page 12**)

As suggested, we have quantitatively analyzed the sediment during the centrifugation step and found that very limited amount of visible agglomerated solid was left in the centrifugation tube (less than ~10 wt% relative to the total solid content).

Response letter Fig. R2.5. (A) The total amount of the graphene products obtained from 15-h milling of 0.1g graphite and 0.4g PEI. (B) The total centrifugation sediment. Inset is the sediment being washed out and filtrated on a nylon membrane for weighting.

Q5. The other evidence of monolayer production is convergent-beam electron diffraction (CBED). The references for this refer to work that had used selected area electron diffraction (SAED) instead. I suggest the authors refer to Latychevskaja et al PNAS 115 (29) DOI: 10.1073/pnas.1722523115. Further details on the measurement is also needed, such as spot-size and convergence angle, as well as an indication on the bright-field image of where the diffraction is acquired from. The two bright-field images in fig S3-3 appear very different, with slightly different magnification, and possibly a different aperture used? Can the authors clarify the reasons for the differences?

Authors: Thank you for your kind suggestion. References for monolayer graphene CBED diffraction pattern has been added in the revised manuscript as suggested. Details of CBED acquisition experiments have been provided in the revised manuscript. Revisions are as follows:

“Fig. 1 Sticky mechanical exfoliation and the physical properties of as-prepared graphene. a, Photograph of the obtained 110-g graphene in powder form. **b,** Powder XRD patterns of pristine graphite and graphene fabricated by 5h, 10h, and 15h milling. **c,** Accumulated Raman spectrum of the 15-h milling graphene deposited on mica. Detailed characterization method can be found in supplementary S2.4. **d,** Selected AFM height profiles of graphene exfoliated by 5h, 10h and 15h milling, respectively (from left to right). **e,** HR-TEM image of exfoliated graphene (inset, atomic resolution). **f,** Converged beam electric diffraction (CBED) of graphene

indicated by the yellow circle in e. Spot size: 9, convergence angle: ~15 milli-rad. g, Diffraction spot intensity taken along the lines in f.” (Main text, page 3)

Figure S3.3A and Figure S3.3B in the previous version of SI were acquired from the same piece of graphene before and after performing CBED acquisition. The difference in morphology is caused by the burning of monolayer graphene under the illumination of electron beam. To clearly visualize this fast burning process, we focused the TEM electron beam on a graphene monolayer, which led to a quick shrinking and distortion of graphene nanosheets in tens of seconds (Fig. R2.6). In the revised SI, we provided low-resolution TEM images to replace figure S3.3 as provided in the response to your Question 1 (Response letter, Page 22).

Supplementary Figure R2.6 Consecutively acquired images of the same individual graphene (from left to right). The burning of graphene nanosheet due to the illumination of electron beam during STEM imaging occurs within a timeframe of several tens of seconds.

Q6. For the XPS analysis, please show the full survey scans as well as the nitrogen peak. Can the authors also please explain the peak shifts seen in the nitrogen peak in fig. 2d? Is this due to charging or a change in chemical environment? Also provide information on the baseline function used to obtain the quantification. Does the nitrogen concentration from XPS agree with that from elemental analysis?

Authors: As suggested, the full XPS survey scans have been provided and shown in Figure S4.3. Quantification analysis has been provided in Table S4.2. The positive shift of in the nitrogen peak for PEI exfoliated graphene relative to pure PEI should be the result of intermolecular electron donation from polymer chains to graphene²⁷. This is also the reason why PEI is a well-known and commonly-used electron donor to functionalize carbon nanotubes²⁸.

The nitrogen concentrations calculated from XPS are in line with the result of elemental analysis for samples with high nitrogen content, such as PEI and pristine exfoliated graphene. Since XPS measurement is not a quantitative method²⁹, it is not a surprise that the XPS results for the defunctionalized samples with ultralow nitrogen content (e.g., thermal-treated graphene) are not in line with elemental analysis (Table S4.2). Therefore, we used elemental analysis results for the quantitative discussion in our revised manuscript.

The new results have been added into the revised manuscript and discussion has also revised accordingly as follows:

S4.2 Removal of residual PEI

Supplementary Fig. S4.3. X-ray photoelectron spectroscopy (XPS) full survey scans of PEI, pristine graphene, HCl-treated graphene, and calcined graphene.

Supplementary Table S4.2. The weight ratio changes of residual PEI after acid and thermal treatment calculated based on the result of elemental analysis and XPS results.

Sample	Elemental analysis		XPS	
	Nitrogen (weight ratio%)	Grafted PEI (wt%)	Nitrogen (atomic %)	Grafted PEI (wt%)
PEI Vis-5	31.95	-	30.75	-
Exfoliated graphene	3.67	11.49	3.53	12.07

HCl-treated graphene	2.01	6.29	2.57	8.79
Thermal-treated graphene	0.08	0.25	0.57	1.85

Following discussion has also been added in revised SI.

Apart from elemental analysis, the removal of PEI functionalities was also investigated by X-ray photoelectron spectroscopy (XPS). Full XPS survey scans of pristine graphene powder, de-functionalized graphene powder, and pure PEI are provided as Fig. S4.3. The corresponding nitrogen atomic percentages are calculated by Thermo Scientific Advantage software using the auto peak fitting function and a standard Shirley background was used for all spectral regions. The nitrogen concentration calculated from XPS is in line with the result of elemental analysis for samples with high nitrogen content such as PEI and pristine exfoliated graphene. Considering XPS measurement is not a quantitative method²⁹, it is not a surprise that the XPS results for the defunctionalized sample with ultralow nitrogen content (e.g., thermal-treated graphene) are not in line with elemental analysis (Table S4.2). Therefore, the result of elemental analysis is used for quantitative discussion in the main text. (**Supplementary, Page 27**)

Corresponding revision has been made in the revised manuscript:

Fig. 2 Properties and potential applications of graphene nanosheets. **a**, Photograph of graphene nanosheets (10h milling) dispersed in solvents including water, ethanol, isopropanol (IPA) and dimethylacetamide (DMAc) at a concentration of 0.5 mg/mL (Top). The weight ratio of remained graphene nanosheets in these solutions after fortnight and 30 days, respectively (Bottom). **b**, TGA analysis of graphite, PEI and exfoliated graphene (10h milling). **c**, Content of residual PEI on as-prepared graphene (10h milling) and de-functionalized graphene by acid washing and incineration under N₂ atmosphere. **d**, High-resolution XPS spectra of N 1s of PEI, as-prepared graphene and de-functionalized graphene. In pure PEI, a sharp peak at 399.8 was recognized as typical groups of -N/NH/NH₂ in PEI. A weak and broad N 1s peak also appears in exfoliated graphene but with a ~0.4 eV positive shift. Considering uncertainties of ±0.2 eV brought by measurement, the positive shift might be caused by electron transfer from the polymer chains of PEI to graphene²⁷. After HCl-treatment, the N 1s peak tends to be weaker and shift more in comparison to pristine graphene, meaning fewer nitrogen remained and greater electron transfer between PEI and graphene.”

Q7. Overall, the presentation of the work could be clearer. In particular that specific steps taken to prepare samples for each characterisation method. It is currently difficult to follow in the extensive supporting information and can be easily clarified to make the paper much easier to read.

Authors: As suggested, we restructured our supporting information as follows. (1) All sample preparation details were provided in one section (section S1.7 Characterization); (2) The discussion of SI has been sorted to adapt to the narrative/discussion order of the manuscript;

(3) Subtitles have been amended or added to help readers navigate through the SI. Please check the revised table of contents of the revised SI:

Contents

S1 Experiment	Error! Bookmark not defined.
S1.1 Materials	Error! Bookmark not defined.
S1.2 Processing parameters.....	Error! Bookmark not defined.
S1.3 Preparation of bulk materials	Error! Bookmark not defined.
S1.4 Preliminary screening.....	Error! Bookmark not defined.
S1.5 Rinsing	Error! Bookmark not defined.
S1.6 Preparation of powder and liquid dispersions.....	Error! Bookmark not defined.
S1.7 Characterizations	Error! Bookmark not defined.
S1.8 Applications.....	Error! Bookmark not defined.
S2 Exfoliation performance	Error! Bookmark not defined.
S2.1 Exfoliation yield.....	Error! Bookmark not defined.
S2.2 Apparent height of graphene nanosheets measured by AFM.....	Error! Bookmark not defined.
S2.3 Lateral size tunability	Error! Bookmark not defined.
S2.4 Further evidence on monolayer graphene	Error! Bookmark not defined.
S2.5 Comparison with other non-chemistry synthesis methods.....	Error! Bookmark not defined.
S3 Practicability	Error! Bookmark not defined.
S3.1 Scale-up study.....	Error! Bookmark not defined.
S3.2 Dispersibility of graphene nanosheets in different solvents	Error! Bookmark not defined.
S3.3 Drying for storage and Re-dispersion.....	Error! Bookmark not defined.
S4 Residual PEI	Error! Bookmark not defined.
S4.1 PEI functionalization.....	Error! Bookmark not defined.
S4.2 Removal of residual PEI.....	Error! Bookmark not defined.
S4.3 Conductivity of graphene nanosheets	Error! Bookmark not defined.
S4.4 Zeta potential of synthesized nanosheets	Error! Bookmark not defined.
S4.5 Ion transport of HCl-treated membranes	Error! Bookmark not defined.
S5 Exfoliation mechanism	Error! Bookmark not defined.
S5.1 Graphene at intermediate stage.....	Error! Bookmark not defined.
S5.2 DEM simulation.....	Error! Bookmark not defined.
S5.2.1 Governing equations.....	Error! Bookmark not defined.
S5.2.2 Parameters for the DEM simulation	Error! Bookmark not defined.
S5.2.3 Results and discussion of DEM simulation.....	Error! Bookmark not defined.
S5.3 Surface morphology characterization of grinding balls	Error! Bookmark not defined.

S5.4 Effect of PEI:graphite ratio	Error! Bookmark not defined.
S5.5 Density Functional Theory (DFT) calculation	Error! Bookmark not defined.
S5.5.1 Calculation methods	Error! Bookmark not defined.
S5.5.2 Results and discussion of DFT	Error! Bookmark not defined.
S5.6 Effect of viscosity of PEI mixtures	Error! Bookmark not defined.
S5.7 Calculation of viscosity threshold for obtaining monolayer graphene.....	Error! Bookmark not defined.
S.6 Universality of sticky mechanical exfoliation	Error! Bookmark not defined.
S6.1 Other layered crystals	Error! Bookmark not defined.
S6.1.1 Exfoliation of TAPB-PDA COF	Error! Bookmark not defined.
S6.1.2 Exfoliation of ZIF-L.....	Error! Bookmark not defined.
S6.1.3 Exfoliation of g-C ₃ N ₄	Error! Bookmark not defined.
S6.2 Exfoliation of BN.....	Error! Bookmark not defined.
S6.3 Comparison with existing methods.....	Error! Bookmark not defined.

Reviewer #3 (Remarks to the Author):

General comments: The paper on Scalable High Yield Exfoliation of Monolayer Nanosheets is a thorough study of polymer protected ball-milling of layered materials. While there are previous reports on the matter, the authors designed a scalable process to achieve a high yield of monolayers.

Authors: We thank the reviewer for highlighting the key innovation of our study on high yield of monolayer nanosheets.

Specific comments:

Q1: From my perspective the conclusions should be taken carefully since for some reason the geometry of the exfoliated crystals seems by SEM (Fig. S3.1) is remarkably different with what was used for the AFM statistics.

Authors: Thanks for pointing out the discrepancy, an issue that is also raised by Reviewer 2. Our further effort proved that this is mainly the result of the observation bias caused by resolution limitation and sample preparation method in SEM characterization. Please refer to our response to Question 1 of Reviewer 2 for the experimental details and our revisions in revised manuscript (**Response letter, Page 21-25**).

Q2: I worry in particular by figure S4.1 where it is clear they are basing statistics in what appears to be solvent or polymer drops. With this in mind I concur that some level of exfoliation was achieved from the Raman data, however I would not call it monolayer judging by the I2D/IG intensity ratio. I also base my conclusion on figure S3.3 that is clearly a bilayer.

Authors: We received the same concern from Reviewer 1 (Question 2) as well. After reviewing and repeating AFM characterizations, we found that the “flower- like” structure in Fig S4.1 was actually a tiny concentrated reservoir of small nanosheets and was caused by the variations of local evaporation rate of sample preparation solution (ethanol based) on mica. We also excluded the possibility of miscounting PEI drops as nanomaterials since PEI presents itself as a highly rough surface with ripples perpendicular to the moving direction of cantilever instead of flat objects (with clear z-direction nanometer steps). These ripples are possibly a result of the strong drag effect of the PEI polymer matrix on the cantilever. As suggested by Reviewer 1 and 2, we coupled Raman with AFM. The regained Raman spectra feature symmetrical 2D bands and the intensity ratios of I(2D)/I(G) locate in the range of 1.06-3.1 (Supplementary Fig. 2.6), consolidating the judgment that these flat discs in AFM are indeed graphene nanosheets and agreeing with the conclusion of high monolayer percentage in the product. Please refer to

our response to Question 2 of Reviewer 1 for all the experimental details and our revisions (**Response letter, Page 4-13**).

Graphene nanosheet in Fig. S3.3 was chosen for acquiring CEBD diffraction patterns that follow the fingerprint of monolayer graphene by appearing them as a single group of diffraction spots with more intense inner spots than outer ones. The reason that Fig 3.3 looks like a bilayer could be the burning of graphene caused by a focused electron beam during the acquisition of CEBD patterns, which caused a quick shrinking and distortion of graphene nanosheets in several tens of seconds. We have demonstrated this burning process by constantly illuminating the same piece of graphene for tens of seconds using TEM beam. Please refer to our response to Question 5 of Reviewer 2 for details as this reviewer had the similar concern about Fig. 3.3 (**Response letter, Page 28-29**).

We also performed low-resolution TEM imaging on our graphene nanosheets and generated their SAED diffraction patterns at the same time. The obtained results provide coherent morphology information with that of AFM results and present additional evidence for confirming monolayer graphene. Please refer to our response to Question 2 of Reviewer 1 for the details (Fig. S2.7, **Response letter, Page 10**).

Q3: Furthermore, I fail to see the reasoning in choosing the other layered materials tested since some of them were prepared in house. I would not call the exfoliation of carbon nitride successful as it is full of defects and, comparing with boron nitride, the problem could have come from the starting material.

Authors: The four different layered materials which are held via different interlayer forces were chosen to validate the universality of the exfoliation method. Covalent organic framework TAPA-PDA, zeolitic imidazolate framework ZIF-L, porous graphitic carbon nitride is held via van de Waals forces and/or π - π interaction. Boron nitride is held via intense polarized interaction. Their shear strengths (the required exfoliation energy calculated by DFT, Fig. 4b) are in a wide range from 0.01 to 0.190 GPa. Although some in-plane defects existed in our homemade starting materials (carbon nitride and ZIF-L), the starting particles were exfoliated into thin nanosheets (0.6-0.8 nm) based on SEM and AFM statistical analysis (Fig. 4, supplementary Figs. S6.2, S6.5). In addition, the crystalline structures of the obtained product were largely preserved as confirmed by powder XRD patterns (supplementary Fig S6.5). Furthermore, the TEM morphologies of the nanosheets present as transparent flat objects with uniform contrast, extended lateral size, and in-plane wrinkles (Figs. 4d-f), which are typical

properties of 2D nanosheets. We believe these results of the four different layered materials will provide new insights to the development of highly efficient exfoliation methods for 2D materials.

References

1. Surekha G, Krishnaiah KV, Ravi N, Suvarna RP. FTIR, Raman and XRD analysis of graphene oxide films prepared by modified Hummers method. In: *Journal of Physics: Conference Series*. IOP Publishing (2020).
2. Dervishi E, *et al.* Large-scale graphene production by RF-cCVD method. *Chem. Commun.*, 4061-4063 (2009).
3. Roddaro S, Pingue P, Piazza V, Pellegrini V, Beltram F. The optical visibility of graphene: interference colors of ultrathin graphite on SiO₂. *Nano Lett.* **7**, 2707-2710 (2007).
4. Li H, *et al.* Rapid and reliable thickness identification of two-dimensional nanosheets using optical microscopy. *ACS nano* **7**, 10344-10353 (2013).
5. Silva DL, *et al.* Raman spectroscopy analysis of number of layers in mass-produced graphene flakes. *Carbon* **161**, 181-189 (2020).
6. Hernandez Y, *et al.* High-yield production of graphene by liquid-phase exfoliation of graphite. *Nature nanotechnology* **3**, 563-568 (2008).
7. Andreeva DV, *et al.* Two-dimensional adaptive membranes with programmable water and ionic channels. *Nat. Nanotechnol.* **16**, 174-180 (2021).
8. Ciesielski A, Samori P. Graphene via sonication assisted liquid-phase exfoliation. *Chem. Soc. Rev.* **43**, 381-398 (2014).
9. Xu K, Cao P, Heath JR. Graphene visualizes the first water adlayers on mica at ambient conditions. *Science* **329**, 1188-1191 (2010).
10. Yuan S, Li Y, Xia Y, Selomulya C, Zhang X. Stable cation-controlled reduced graphene oxide membranes for improved NaCl rejection. *J. Membr. Sci.* **621**, 118995 (2021).
11. Paton KR, *et al.* Scalable production of large quantities of defect-free few-layer graphene by shear exfoliation in liquids. *Nat. Mater.* **13**, 624-630 (2014).
12. Latychevskaia T, *et al.* Convergent beam electron holography for analysis of van der Waals heterostructures. *PNAS* **115**, 7473-7478 (2018).
13. Soule D, Nezbeda C. Direct basal-plane shear in single-crystal graphite. *J. Appl. Phys.* **39**, 5122-5139 (1968).
14. Rooney A, *et al.* Anomalous twin boundaries in two dimensional materials. *Nat. Commun.* **9**, 1-7 (2018).

15. Li Z, *et al.* Mechanisms of liquid-phase exfoliation for the production of graphene. *ACS nano* **14**, 10976-10985 (2020).
16. Lei W, Mochalin VN, Liu D, Qin S, Gogotsi Y, Chen Y. Boron nitride colloidal solutions, ultralight aerogels and freestanding membranes through one-step exfoliation and functionalization. *Nat. Commun.* **6**, 1-8 (2015).
17. Chen S, *et al.* Simultaneous production and functionalization of boron nitride nanosheets by sugar-assisted mechanochemical exfoliation. *Adv. Mater.* **31**, 1804810 (2019).
18. Zhang L, *et al.* Solid Phase Exfoliation for Producing Dispersible Transition Metal Dichalcogenides Nanosheets. *Adv. Funct. Mater.* **30**, 2004139 (2020).
19. Ji J, *et al.* Simultaneous noncovalent modification and exfoliation of 2D carbon nitride for enhanced electrochemiluminescent biosensing. *Journal of the American Chemical Society* **139**, 11698-11701 (2017).
20. McManus D, *et al.* Water-based and biocompatible 2D crystal inks for all-inkjet-printed heterostructures. *Nat. Nanotechnol.* **12**, 343-350 (2017).
21. Coleman JN, *et al.* Two-dimensional nanosheets produced by liquid exfoliation of layered materials. *Science* **331**, 568-571 (2011).
22. Zhang Z, *et al.* Bioinspired graphene oxide membranes with pH-responsive nanochannels for high-performance nanofiltration. *ACS nano* **15**, 13178-13187 (2021).
23. Kang Y, Xia Y, Wang H, Zhang X. 2D laminar membranes for selective water and ion transport. *Adv. Funct. Mater.* **29**, 1902014 (2019).
24. Zhang M, Zhao P, Li P, Ji Y, Liu G, Jin W. Designing Biomimic Two-Dimensional Ionic Transport Channels for Efficient Ion Sieving. *ACS nano* **15**, 5209-5220 (2021).
25. Zhan H, Xiong Z, Cheng C, Liang Q, Liu JZ, Li D. Solvation - Involved Nanoionics: New Opportunities from 2D Nanomaterial Laminar Membranes. *Adv. Mater.* **32**, 1904562 (2020).
26. Shen J, Liu G, Han Y, Jin W. Artificial channels for confined mass transport at the sub-nanometre scale. *Nat. Rev. Mater.*, 1-19 (2021).
27. Zeng X, *et al.* Simultaneously tuning charge separation and oxygen reduction pathway on graphitic carbon nitride by polyethylenimine for boosted photocatalytic hydrogen peroxide production. *ACS Catal.* **10**, 3697-3706 (2020).

28. Mo C, *et al.* Boosting water oxidation on metal-free carbon nanotubes via directional interfacial charge-transfer induced by an adsorbed polyelectrolyte. *Energy & Environmental Science* **11**, 3334-3341 (2018).

29. Beccat P, Da Silva P, Huiban Y, Kasztelan S. Quantitative surface analysis by XPS: application to hydro-treating catalysts; Analyse quantitative de surface par XPS (X-ray photoelectron spectroscopy): application aux catalyseurs d'hydrotraitement. *Oil & Gas Science and Technology* **54**, (1999).

REVIEWER COMMENTS

Reviewer #1 (Remarks to the Author):

The authors have thoroughly revised their manuscript and addressed all questions raised. It appears there might have initially been issues with AFM sample preparation and imaging so that it is hard to tell whether all images show a reliable sample morphology. Regardless of this, the Raman and diffraction data is a convincing demonstration that monolayers of graphene were indeed produced. Publication is now recommended.

Reviewer #2 (Remarks to the Author):

The authors have revised the manuscript and supporting information to address the concerns raised in my previous review. These have mostly improved the manuscript, however there are some areas where I continue to have concerns.

1. The authors have removed the SEM images, as they suggest that there was inadvertent selection of flakes occurring during the sample preparation for this imaging. I appreciate the authors have recognised this as a potential issue. However, the AFM images presented do not show any of the large, straight-edged particles that were shown in the original manuscript. If they were not present, then it would be good to know why not. Otherwise, then images of these particles need to be included to ensure a representative presentation of the flakes produced is given to readers.

2. With respect to the XPS analysis, the authors have said the XPS is not quantitative, however none-the-less they present quantitative information is presented. I would firstly suggest that XPS is indeed quantitative, with a large body of literature over many years confirming this. See for example Practical guides for x-ray photoelectron spectroscopy: Quantitative XPS: Journal of Vacuum Science & Technology A: Vol 38, No 4 (2020), Quantitative XPS: I. Analysis of X-ray photoelectron intensities from elemental data in a digital photoelectron database - Journal of Electron Spectroscopy and Related Phenomena: Vol 120. No 1-3 (2001) and The quantitative analysis of surfaces by XPS: A review: Surface and Interface Analysis; Vol 2 (6) (1980). I suspect the authors mean that the concentration of nitrogen is below the limit of accurate quantification, which may be correct. However, the value reported by the authors is above the expected detection limit for nitrogen in a carbon matrix. See: Detection limits in XPS for more than 6000 binary systems using Al and Mg K α X-rays - Surface Interface and Analysis. Vol 46 No 3 (2014).

3. The UV-vis extinction spectrum shown in Fig. R2.1A shows an unexpected feature at \sim 260 nm, which is not seen for the filtered solution. What is the origin of this feature, and does it affect the conclusions drawn about the loss of material through the membrane?

4. When describing the centrifuge conditions, please give the speed in RCF rather than rpm, to allow others to replicate the work accurately. At the very least, include the rotor used as well as the centrifuge model.

5. The Raman of the small, irregular features seen in AFM is welcome, but care needs to be taken interpreting the results. Given the AFM images of the measured areas, then multiple flakes are certainly being illuminated in a single measurement. As such, the 2D peak is likely to be affected, and so the width of this peak should be checked to confirm true monolayer graphene present. I am also curious about the differing signal-to-noise ratio in the three spectra presented. Has there been a loss of focus between measurement locations? Overall, the D-band height is lower than I would expect from such small lateral sized flakes.

Reviewer #3 (Remarks to the Author):

The authors thoroughly addressed the numerous concerns raised by the reviewers.

Response Letter

Scalable High Yield Exfoliation for Monolayer Nanosheets

(NCOMMS-22-13685-A)

REVIEWER COMMENTS

Reviewer #1 (Remarks to the Author):

General comments: The authors have thoroughly revised their manuscript and addressed all questions raised. It appears there might have initially been issues with AFM sample preparation and imaging so that it is hard to tell whether all images show a reliable sample morphology. Regardless of this, the Raman and diffraction data is a convincing demonstration that monolayers of graphene were indeed produced. Publication is now recommended.

Authors: We thank the reviewer for the positive recommendation.

Reviewer #2 (Remarks to the Author):

General comments: The authors have revised the manuscript and supporting information to address the concerns raised in my previous review. These have mostly improved the manuscript, however there are some areas where I continue to have concerns.

Authors: We thank the reviewer for acknowledging the significant improvement of the manuscript. The remaining concerns have been further addressed in the revised manuscript as detailed below.

Specific comments: Q1. The authors have removed the SEM images, as they suggest that there was inadvertent selection of flakes occurring during the sample preparation for this imaging. I appreciate the authors have recognised this as a potential issue. However, the AFM images presented do not show any of the large, straight-edged particles that were shown in the original manuscript. If they were not present, then it would be good to know why not. Otherwise, then images of these particles need to be included to ensure a representative presentation of the flakes produced is given to readers.

Authors: Thank the reviewer for the valuable suggestion. Representative AFM images of large, sharp-edged multilayer nanosheets were provided in the revised supporting information (Fig. S2.8C), together with SEM broad observation and TEM diffraction pattern (Figs. S2.8A, B) as follows.

S2.4 Analysis on large multilayers

Supplementary Fig. S2.8. (A) SEM image of graphene nanosheets (15-h milling) deposited on a porous AAO disc. (B) TEM image of a multilayer graphene and its SAED diffraction pattern. (C) AFM topological image and height profile of multilayer graphene nanosheets.

When graphene nanosheets were deposited on a porous AAO disc, large nanosheets in the middle (yellow cycle) displaying a clear shape were observed. Although some small nanosheets can pass through the AAO disc that has a pore size similar to theirs, upon close observation, we still found many small nanosheets with round shapes and serrated edges. The morphologies of the small nanosheets agree with AFM observation (Fig. S2.3), which are confirmed as monolayer graphene by Raman spectra and SEM diffraction patterns (Figs. S2.6, S2.7). By contrast, large nanosheets with sharp edges (Fig. S2.8A) are possibly either the stack of several nanosheets with self-folding or multi-layers that have not been fully exfoliated, which are evidenced by TEM diffraction patterns and AFM height profiles (Figs. S2.8B, C).

The morphology difference between monolayer and multilayer nanosheets could be the result of the in-plane breaking processes. Initial delamination and breaking start from structural deformation at mechanically vulnerable crystalline structural deformations on material surface, often named “kind band striations”^{1, 2, 3}. This crystalline structural breaking leads to incompletely exfoliated few-layer nanosheets with large sizes and sharp edges. The large nanosheets then undergo further in-plane breaking by collisions of protruding sharp ridges on the grinding ball surface (Fig. 3b), which provides sufficiently high compression to break graphene nanosheets at any point due to concentrated forces. Considering the highly random

distribution of these ridges over the balls, the long edges of the nanosheets have a better chance to be trimmed than short edges during milling, as a result, smaller nanosheets that have undergone more times of breaking tend to be more likely to end up with higher roundness but with serrated edges. (Supplementary Page 18-19)

Q2. With respect to the XPS analysis, the authors have said the XPS is not quantitative, however none-the-less they present quantitative information is presented. I would firstly suggest that XPS is indeed quantitative, with a large body of literature over many years confirming this. See for example Practical guides for x-ray photoelectron spectroscopy: Quantitative XPS: Journal of Vacuum Science & Technology A: Vol 38, No 4 (2020), Quantitative XPS: I. Analysis of X-ray photoelectron intensities from elemental data in a digital photoelectron database - Journal of Electron Spectroscopy and Related Phenomena: Vol 120. No 1-3 (2001) and The quantitative analysis of surfaces by XPS: A review: Surface and Interface Analysis; Vol 2 (6) (1980). I suspect the authors mean that the concentration of nitrogen is below the limit of accurate quantification, which may be correct. However, the value reported by the authors is above the expected detection limit for nitrogen in a carbon matrix. See: Detection limits in XPS for more than 6000 binary systems using Al and Mg K α X-rays – Surface Interface and Analysis. Vol 46 No 3 (2014).

Authors: We thank the reviewer for the detailed explanation. Considering both the elemental analysis CHNS and the XPS can provide useful quantitative information, in the revised SI, we report all the results to provide better reference to readers. The results of CHNS and the XPS quantitative analysis are in line with each other for samples with high nitrogen content. However, the deviations between them become large when the nitrogen content approaches the detection limit. We believe that the difference between the results of the two analysis methods at low nitrogen content stems from the different sample preparation procedures and characterization mechanisms. For CHNS analysis, milligram of graphene powder samples is completely oxidized by “flash combustion”. The combustion products are separated and analyzed to give signals proportional to the concentration of individual components including carbon, hydrogen, nitrogen, sulphur and oxygen of the bulk samples⁴. For XPS analysis, graphene powder was dispersed in ethanol and filtrated onto nylon membranes. Only the top surface is analyzed, as a result, the result of XPS can easily deviate at low element content from CHNS analysis which analyzes whole sample^{5, 6}.

Corresponding revisions have been made in the revised SI.

The nitrogen concentration calculated from XPS is in line with the result of elemental analysis (CHNS) for samples with high nitrogen content such as PEI and pristine exfoliated graphene (Table S4.2). However, the difference between the results of the XPS and CHNS analysis rises with the nitrogen contents in the samples approach the detection limit, especially for the thermal-treated graphene. The difference between the results of the two methods may stems from the difference of their sample preparation methods and analyzing mechanisms. For the CHNS analysis, bulk powder was completely oxidized by “flash combustions”. The combustion products are separated and analyzed to give quantitative information on components in the samples⁴. By comparison, graphene was deposited on a substrate and shot by the X-rays for the XPS analysis. The qualitative and quantitative signals are only generated and analyzed from the sample surface^{5, 6}. In summary, the PEI removal efficiency of acid washing is at 45.3%, and 27.2% measured by the CHNS and XPS analysis, respectively, which can be further increased to 97.8% (CHNS) / 84.7% (XPS) by applying thermal treatment. (Supplementary Page 28)

Q3. The UV-vis extinction spectrum shown in Fig. R2.1A shows an unexpected feature at ~260 nm, which is not seen for the filtered solution. What is the origin of this feature, and does it affect the conclusions drawn about the loss of material through the membrane?

Authors: We thank the reviewer for pointing out this detail. The UV-Vis peak at ~260 is attributed to the grafted PEI functionalities since no such a peak can be identified from the totally de-functionalized graphene sample (Fig. R1).

Response letter Fig. R1. UV-Vis spectrum of calcined graphene water solution at a concentration of 10 $\mu\text{g/mL}$. The graphene nanosheets obtained by 15h milling were freeze-dried into powder, then underwent thermal treatment to remove PEI functionalities, and finally re-dispersed in water by 30-min sonication. Note that the as-prepared water dispersion was characterized by UV-Vis spectroscopy directly without centrifugation.

When we zoomed in Fig. R2.1A (Fig. R2A in this response letter), no obvious peak at ~ 260 nm was observed for either feed solution or filtrated solution. We believe that this is because graphene concentration in the solutions is very low. As shown in Fig. R2B, there is a decent linear relationship between the graphene concentration and the UV-Vis absorbance at 260 nm. According to the fitted curve (Fig. R2B) and the absorbance of the two solutions at 260 nm (Fig. R2A), the graphene concentrations of the feed solution and the filtrated solution in Fig. are calculated to be 0.24 $\mu\text{g/mL}$ and 0.086 $\mu\text{g/mL}$, respectively. The results indicate that more than one third of graphene nanosheets passed through the AAO membrane. This quantitative data further supports our conclusion that graphene nanosheets were lost during our SEM sample preparation.

Response letter Fig. R2. (A) The UV-Vis spectra of previous Fig. R2.1A, inset is the spectra zoomed in at around 260. (B) Calibration standard curve of the absorbance of graphene nanosheet dispersions at 260 nm as a function of their concentrations. This figure is based on the data of UV-Vis results in Supplementary section 2.1.

Q4. When describing the centrifuge conditions, please give the speed in RCF rather than rpm, to allow others to replicate the work accurately. At the very least, include the rotor used as well as the centrifuge model.

Authors: Thank you for pointing out the problem. Centrifugation details have been provided as suggested.

The obtained nanosheets solution was repeatedly washed to remove excess PEI (Supplementary section S1.5) and centrifugated at RCF of 236 g (Rotor 12181, Sigma 2-16P) for 20 min to remove thick flakes. (Main text Page 12)

Q5a. The Raman of the small, irregular features seen in AFM is welcome, but care needs to be taken interpreting the results. Given the AFM images of the measured areas, then multiple flakes are certainly being illuminated in a single measurement. As such, the 2D peak is likely to be affected, and so the width of this peak should be checked to confirm true monolayer graphene present.

Authors: We thank the reviewer for the suggestion. Although the 2D band becomes broader with the increase of graphene thickness, the difference between their width is not obvious enough to provide solid evidence. For instance, a previous study compared graphene samples with thickness increase from single to 19 layers⁷, and the results showed that the width of the 2D band increased a little from 1 layers to 2 layers, but there was no apparent difference with further thickness increase. Therefore, we focused on the shape of the 2D band instead of width as additional evidence for monolayer graphene. Since the 2D band originates from a two-phonon double resonance process that is related to the in-plane band structure, a symmetric shape of the 2D band is the critical characteristic for single-layer graphene^{8,9}, which is in line with our case. By contrast, with the dispersion of π electrons from stacked sheets, multilayer graphene shows an asymmetric 2D band with a clear shoulder⁹. Additionally, coupling with the relative intensity of 2D and G bands, the 2D band shapes should provide more reliable evidence from the Raman spectra. The Raman intensity ratio of $I_{2D}/I_G > 1$ ⁷ in our case is in line with the characteristic of monolayer graphene (Supplementary Fig, S2.6C).

We agree with the reviewer that illuminating multiple small graphene nanosheets collectively and the semi in-situ analysis can compromise the reliability of the Raman results. But, coupling the Raman data with other more convincing evidences, e.g., the sub-nm thickness (0.5-0.8 nm) of graphene (Fig. 1d), unimodal thickness distributions (Supplementary Fig. S2.3), and the TEM diffraction patterns (SAED & CBED) (Figs. 1e-g, Supplementary Fig. S2.7), we can carefully conclude that high-percentage monolayer graphene nanosheets present in our products.

Q5b: I am also curious about the differing signal-to-noise ratio in the three spectra presented. Has there been a loss of focus between measurement locations?

Authors: We believe that the different signal-to-noise ratio originated from the difference of sample scanning time between these measurements. To avoid burning nanosheets during characterization, we applied a short integration time of 1s and a low laser power of 5 mW, but scanned repeatedly at the same position to accumulate the signals until the D, G and 2D bands became identifiable. **Position 1** had a relatively more intensive signal in each scan, so it was scanned 50 times for us to collect the data. **Position 2** and **3** were scanned 100 times. The fewer scanning times in **position 1** led to a higher signal-to-noise ratio of the overall accumulated result. Details have been added in the figure caption as follows.

“Experimental conditions: integration time 1s, laser: 532-nm with a power of 5 mW, number of scans: 50 for position 1, 100 for position 2 and 3.” (Supplementary Page 17)

Q5c: Overall, the D-band height is lower than I would expect from such small lateral sized flakes.

Authors: The D-band is due to the basal plane defects and edge defects¹⁰. Given the edge defects are inevitable and affected by the size, the low D-band here could be the result of low vacancy defects in this “polymer-protected” ball-milling derived graphene. Previous research suggests that the relative intensity of D-band is largely influenced by the defect types. The relative intensity of D band follows the sequence of graphene with sp^3 defects (oxidation) > with vacancy defects > with boundary defects¹¹. Because of the simultaneous breaking and mechanochemical functionalization process along the edges, ball milling process has been proven to be a convenient and highly selective technology for the edge functionalization of 2D materials^{12, 13, 14, 15}. As a result, the low D bands in Raman spectra could come from the limited in-plane vacancy and oxidation defects, but dominant edge defects of our graphene. To confirm our hypothesis, we reviewed the reported works and found the D-band intensity in our work (relative to G-band, $I_D/I_G=0.3-0.5$) is comparable to that of other similar reported ball-milling exfoliation protocols (0.02-0.5)^{16, 17} and high-speed mixing delivered vacancy-defect-free graphene (0.17-0.37)¹⁸, while it is lower than that of graphene oxide (GO) and reduced GO (rGO) (~0.7-0.9)¹⁹ with in-plane vacancy or oxidation defects.

A brief discussion on the D band intensity has been added in the revised SI:

Furthermore, the $I(D)/I(G)$ band intensity, which is related to defects, is in the range of 0.3-0.5. Such a low I_D/I_G ratio originated from the quality lattice is much lower than that of graphene

oxide (GO) and reduced GO (rGO) (~0.7-0.9)¹⁹, but it is comparable to high-speed mixing delivered vacancy-defect-free few-layer graphene (0.17-0.37)¹⁸. (**Supplementary Page 18**)

Reviewer #3 (Remarks to the Author):

General comments: The authors thoroughly addressed the numerous concerns raised by the reviewers.

Authors: Thanks for the reviewer's acknowledgment and recommendation.

Reference

1. Soule D, Nezbeda C. Direct basal-plane shear in single-crystal graphite. *J. Appl. Phys.* **39**, 5122-5139 (1968).
2. Rooney A, *et al.* Anomalous twin boundaries in two dimensional materials. *Nat. Commun.* **9**, 1-7 (2018).
3. Li Z, *et al.* Mechanisms of liquid-phase exfoliation for the production of graphene. *ACS nano* **14**, 10976-10985 (2020).
4. Fadeeva V, Tikhova V, Nikulicheva O. Elemental analysis of organic compounds with the use of automated CHNS analyzers. *J. Anal. Chem.* **63**, 1094-1106 (2008).
5. Seah M. The quantitative analysis of surfaces by XPS: a review. *Surf. Interface Anal.* **2**, 222-239 (1980).
6. Beccat P, Da Silva P, Huiban Y, Kasztelan S. Quantitative surface analysis by XPS: application to hydro-treating catalysts; Analyse quantitative de surface par XPS (X-ray photoelectron spectroscopy): application aux catalyseurs d'hydrotraitement. *Oil & Gas Science and Technology* **54**, (1999).
7. Silva DL, *et al.* Raman spectroscopy analysis of number of layers in mass-produced graphene flakes. *Carbon* **161**, 181-189 (2020).
8. Ni Z, Wang Y, Yu T, Shen Z. Raman spectroscopy and imaging of graphene. *Nano Research* **1**, 273-291 (2008).
9. Tang B, Guoxin H, Gao H. Raman spectroscopic characterization of graphene. *Applied Spectroscopy Reviews* **45**, 369-407 (2010).

10. Hernandez Y, *et al.* High-yield production of graphene by liquid-phase exfoliation of graphite. *Nat. Nanotechnol.* **3**, 563-568 (2008).
11. Eckmann A, *et al.* Probing the nature of defects in graphene by Raman spectroscopy. *Nano Lett.* **12**, 3925-3930 (2012).
12. Jeon I-Y, *et al.* Large-scale production of edge-selectively functionalized graphene nanoplatelets via ball milling and their use as metal-free electrocatalysts for oxygen reduction reaction. *Journal of the American Chemical Society* **135**, 1386-1393 (2013).
13. Ding J, Zhao H, Yu H. Graphene nanofluids based on one-step exfoliation and edge-functionalization. *Carbon* **171**, 29-35 (2021).
14. Xiang Z, Dai Q, Chen JF, Dai L. Edge functionalization of graphene and two-dimensional covalent organic polymers for energy conversion and storage. *Adv. Mater.* **28**, 6253-6261 (2016).
15. Jeon IY, Bae SY, Seo JM, Baek JB. Scalable production of edge-functionalized graphene nanoplatelets via mechanochemical ball-milling. *Adv. Funct. Mater.* **25**, 6961-6975 (2015).
16. González-Domínguez JM, León V, Lucío MI, Prato M, Vázquez E. Production of ready-to-use few-layer graphene in aqueous suspensions. *Nature protocols* **13**, 495 (2018).
17. Yang L, *et al.* Glue-assisted grinding exfoliation of large-size 2D materials for insulating thermal conduction and large-current-density hydrogen evolution. *Mater. Today* **51**, 145-154 (2021).
18. Paton KR, *et al.* Scalable production of large quantities of defect-free few-layer graphene by shear exfoliation in liquids. *Nat. Mater.* **13**, 624-630 (2014).
19. Yuan S, Li Y, Xia Y, Selomulya C, Zhang X. Stable cation-controlled reduced graphene oxide membranes for improved NaCl rejection. *J. Membr. Sci.* **621**, 118995 (2021).

REVIEWERS' COMMENTS

Reviewer #2 (Remarks to the Author):

The authors have addressed my comments sufficiently to recommend publication.